# Deep-sea hiatus record reveals orbital pacing by 2.4 Myr eccentricity grand cycles

Adriana Dutkiewicz [1] ✉, Slah Boulila[2,3] & R. Dietmar Müller [1]

Astronomical forcing of Earth's climate is embedded in the rhythms of stratigraphic records, most famously as short-period ($10^4$–$10^5$ year) Milankovitch cycles. Astronomical grand cycles with periods of millions of years also modulate climate variability but have been detected in relatively few proxy records. Here, we apply spectral analysis to a dataset of Cenozoic deep-sea hiatuses to reveal a ~2.4 Myr eccentricity signal, disrupted by episodes of major tectonic forcing. We propose that maxima in the hiatus cycles correspond to orbitally-forced intensification of deep-water circulation and erosive bottom current activity, linked to eccentricity maxima and peaks in insolation and seasonality. A prominent episode of cyclicity disturbance coincides with the Paleocene-Eocene Thermal Maximum (PETM) at ~56 Myr ago, and correlates with a chaotic orbital transition in the Solar System evident in several astronomical solutions. This hints at a potential intriguing coupling between the PETM and Solar System chaos.

In 1976 Hays et al.[1] demonstrated for the first time the presence of $10^4$–$10^5$ year astronomical cycles in Pleistocene deep-sea sediments, confirming Milankovitch's theory that Earth's climate is modulated by periodicities in perturbations of Earth's orbit around the Sun and Earth's spin axis[2]. Apart from the well-known astronomical cycles with periods of 19 kyr, 23 kyr, 41 kyr, 100 kyr, and 400 kyr that pace Earth's climate[3], the geological record also contains signals of much longer-period "grand cycles"[4], which are predicted by astronomical theory[5]. These "grand cycles" include orbitally-forced periodicities of millions and even tens of millions of years that are similarly linked to changes in incoming solar radiation and paleoclimate[6–9]. The 2.4 Myr (g4–g3) eccentricity cycle related to the precession of the perihelions of Earth (g3) and Mars (g4), and the 1.2 Myr (s4–s3) obliquity cycle associated with the precession of the nodes of the two planets, are of particular interest[5]. Retrieving these signals from the geological record can provide critical information on Earth-Mars secular resonance and the timing of chaotic episodes in the inner Solar System, potentially reducing uncertainties in the computations of Earth's orbital motion before 50 Ma, and extending astronomical calibrations of the geological timescale[6].

Most studies of grand orbital cycles are focused on relatively short (<10 Myr), high-resolution continuous stratigraphic records that yield several eccentricity cycles in the stable 405 kyr band[10,11], but very few longer-period (2.4 Myr) modulating cycles e.g., ref. [12]. The ~2.4 Myr eccentricity cycle, however, has been found embedded in the Cenozoic $\delta^{18}O$ and $\delta^{13}C$ isotope record[7–9,12,13], in geophysical signals of stratigraphic sequences[4,14–16], and in fossil assemblages[17,18]. The cycles are variably attributed to changes in temperature and ice-volume[7], fluctuations in sediment and organic carbon accumulation linked to river runoff[8], variations in seawater temperature[12], and changes in ocean circulation and structure[17] − all of which are driven by astronomically forced changes in insolation and climate. Here we take a different approach by using the very aspect of stratigraphy that is the bane of cyclostratigraphy − stratigraphic discontinuities. We use a merged record of 370 deep-sea hiatuses (breaks in sedimentation) from the global ocean spanning 70–0 Ma based on the compilation of Dutkiewicz and Müller[19] from 293 scientific deep-sea drill holes (Fig. 1; see "Methods" section). Our analysis shows the presence of ~2.4 Myr eccentricity cycles linked to orbitally-paced erosion of the seafloor by deep-sea currents, as well as a possible astronomical chaotic transition between ~56 and 53 Ma.

[1]EarthByte Group, School of Geosciences, The University of Sydney, Sydney, NSW 2006, Australia. [2]Sorbonne Université, CNRS, Institut des Sciences de la Terre Paris, ISTeP, 75005 Paris, France. [3]ASD/IMCCE, CNRS-UMR8028, Observatoire de Paris, PSL University, Sorbonne Université, 75014 Paris, France. ✉e-mail: adriana.dutkiewicz@sydney.edu.au

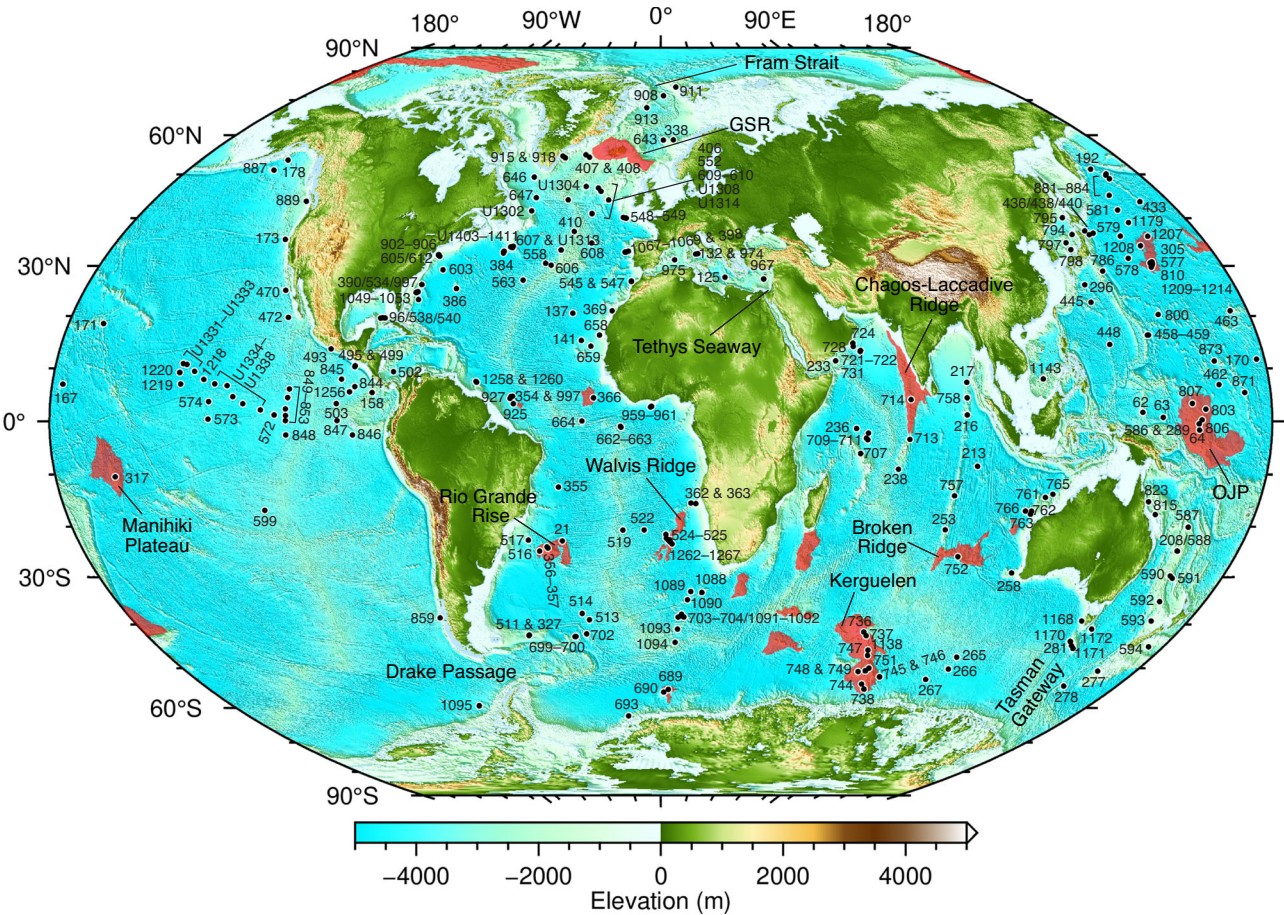

**Fig. 1 | Location map of drill sites.** Black circles and numbers denote scientific ocean drilling sites of the Deep-Sea Drilling Project, Ocean Drilling Program, Integrated Ocean Drilling Program, and the International Ocean Discovery Program used in this study. Red regions indicate major large igneous provinces (LIPs) on oceanic crust following Prokoph et al.[70]. Note ocean gateways mentioned in the text. GSR − Greenland-Scotland Ridge, OJP − Ontong-Java Plateau. Elevation from ETOPO1 global relief model (https://www.ngdc.noaa.gov/mgg/global/). Winkel Tripel projection.

## Results and discussion

### Hiatus cyclicity during the Cenozoic

We used a weighted-average low-pass filter to remove long-term variations in the hiatus-frequency time series (Fig. 2a), and to highlight the short-term regular oscillations (Fig. 2b). Spectral analysis and band-pass filtering of the hiatus data illustrates that the dominant hiatus-frequency cycles in this time series have periods of 2–3 Myr (Fig. 2c). However, the cycles are not equally pronounced in terms of their amplitude at all times (Fig. 2c). An evolutive Fourier transform analysis reveals four periods of relatively stable unimodal grand cycle occurrence in the data, spanning ~70–56 Ma, ~50–34 Ma, ~30–22 Ma and ~15–0 Ma (Fig. 2d). Spectral analysis per intervals (Fig. 3) shows dominant cycles of 2.34 Myr (70–57 Ma), 2.31 Myr (50–32 Ma), 3.15 Myr (30–20 Ma) and 2.56 Myr (15–0 Ma). The hiatus record is punctuated by three bifurcations with onsets at ~56 Ma, ~34 Ma, and ~22 Ma, during which the dominant cycle frequency splits into two modes (Fig. 2d) with reduced amplitudes (Fig. 2c).

We first explore the relationship between the deep-sea hiatus record and Earth's orbital eccentricity for the interval spanning 50–32 Ma using the theoretical astronomical solutions La2004[5], La2010a–d, and La2011[6,20] describing Earth's orbital motion through time. We choose this interval because it exhibits a continuous sequence of 2.31 Myr hiatus cycles (Fig. 4a) that are very close to the 2.4 Myr (g4−g3) eccentricity cycle band[5,6]. This critical time interval is also within the validity limits of the orbital solutions, which are uncertain beyond ~40–50 Ma due to the chaotic behavior of the Solar System[5,6,20]. La2004 is precise back to 40 Ma[5], while La2010a–d and

La2011 extend the accuracy of La2004 orbital eccentricity model back from 40 Ma to about 50 Ma[6,20]. Nevertheless, the accuracy of La2004, La2010a−d, and La2011 over the interval 50–40 Ma should be potentially explored by correlations with the geological records[6,20]. We find that for the interval 40–32 Ma the hiatus cycles are slightly phase-shifted with the 2.4 Myr cycle band in all astronomical solutions (see "Methods" section). The phase shift significantly increases in the interval 50–40 Ma for all models except for La2004 (Fig. 4). La2004 cycles are intriguingly in phase with hiatus cycles at ~50–46 Ma. These correlations support the idea that the astronomical models with the exception of La2004 lose precision starting at ~40–45 Ma[5,6,20,21]. The match between La2004 and hiatus cycles supports the accuracy of the La2004 model for longer timescale variations[22], and agrees with previous investigations suggesting that the La2004 solution provides a better match with some geological data[9,14,22,23]. We also note that the 2.4 Myr cycles in La2004 and La2010a models are in phase from 44 to 0 Ma. La2010a and La2010b start to diverge significantly at ~48 Ma. Despite La2010b including the effects from five major asteroids, in contrast to La2010c (see "Methods" section), the 2.4 Myr cycles in La2010b and La2010c are in phase over the entire 50–32 Ma time interval. Similarly, La2010d and La2011 2.4 Myr cycles are almost in phase over the whole 50–32 Ma time interval, but both models include the five major asteroids. From the above analyses of various astronomical solutions, it follows that the critical parameters that lead to discrepancies among solutions beyond 40 Ma are the initial conditions (e.g., La2004 vs La2010a−d), followed by the step size of the numerical integration (e.g., La2010a vs La2010b). We use phase and coherence

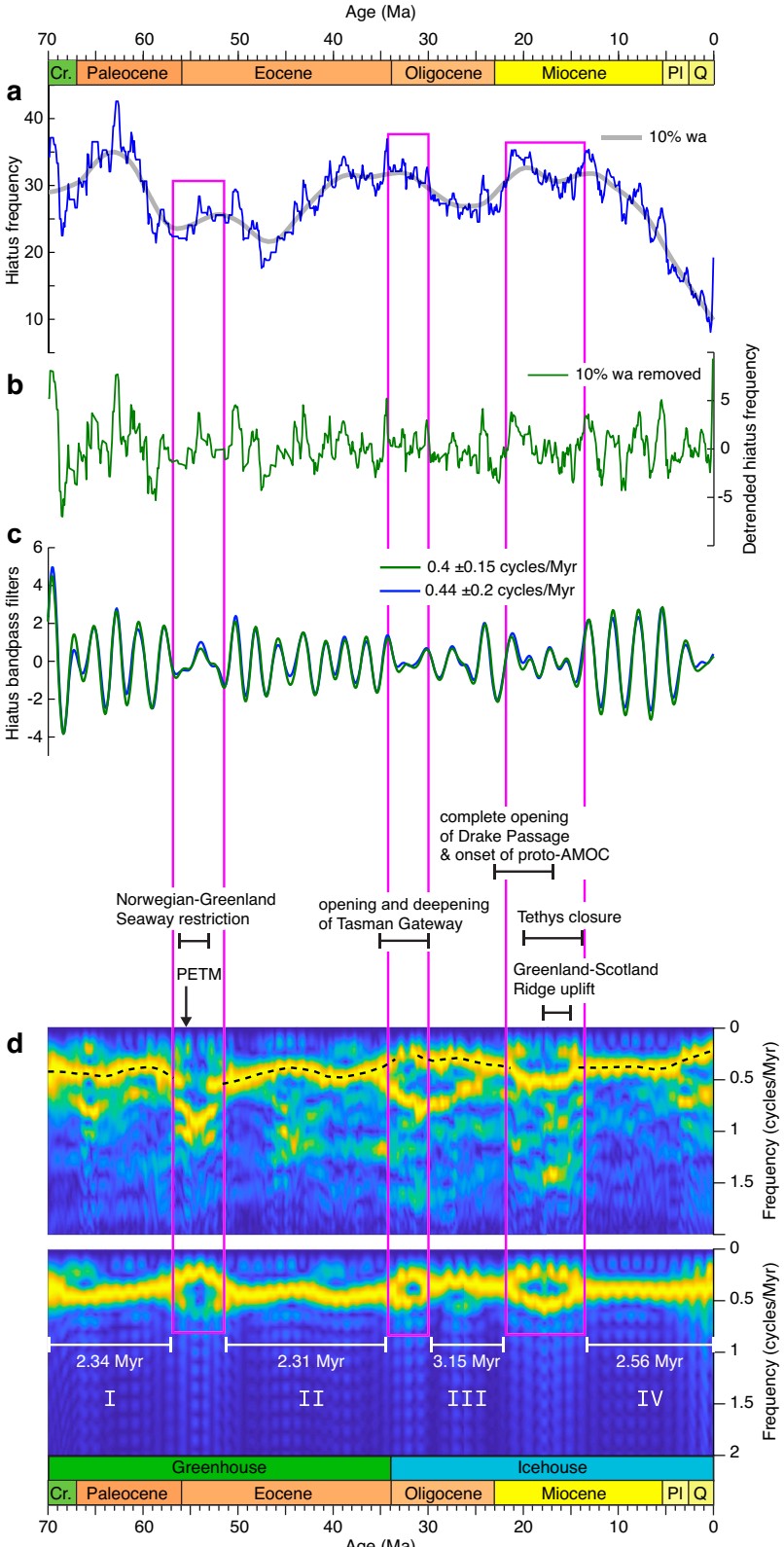

analysis to explore to what extent the hiatus dataset and these different eccentricity model predictions are in phase. We find a strong coherence between the two signals of 0.83–0.93 for cycles at 2.4 Myr, confirming the common 2.4 Myr cyclicity among geological and astronomical signals (Fig. 5).

The vast majority of Cenozoic hiatuses were formed by erosion and redistribution of sediments by bottom currents[19,24,25]. An

assessment of the paleo-water depth at which each hiatus in our dataset formed versus regional reconstructions of the carbonate compensation depth (CCD) – the depth at which the rate of supply of carbonate is balanced by its dissolution[26], shows that only a small number of hiatuses from a dozen holes may either be the result of carbonate dissolution or non-deposition of carbonate[19]. Dissolution events are difficult to detect stratigraphically because they can be

**Fig. 2 | Time-series analysis of global hiatus-frequency data.** Magenta boxes indicate bifurcation intervals. **a** Raw hiatus-frequency data from Dutkiewicz and Müller[19] sampled at 100 kyr intervals (blue), along with the smoothed data based on a 10% weighted average (wa) of the series (gray) for detrending. **b** Detrended hiatus-frequency data with removal of the trend shown in (**a**). **c** Bandpass filters of the 2.4 Myr eccentricity cycle. Note that cycle amplitudes are significantly reduced during bifurcation intervals. **d** Evolutive Fast Fourier Transform (FFT) amplitude spectrograms of the hiatus data. The upper panel shows a spectrogram of a larger frequency band excluding lower frequencies (<0.25 cycles/Myr), retaining higher frequencies up to 1.8 cycles/Myr (window = 6 Myr, step = 0.1 Myr). The lower panel shows a spectrogram of 0.4 ± 0.15 cycles/Myr bandpass filtered hiatus data (window = 6 Myr, step = 0.1 Myr), highlighting the frequencies of interest for the hiatus analysis. White bars labeled I–IV indicate intervals of stable unimodal signal dominated by hiatus cycles ranging from 2.31 Myr to 3.15 Myr over the last 70 Myr (see Fig. 3), separated by bifurcations. Changes in ocean gateways from Hovikoski et al.[45], Eagles and Jokat[50], Straume et al.[49] and Bialik et al.[51]. PETM−Paleocene-Eocene Thermal Maximum, AMOC−Atlantic Meridional Overturning Circulation, Cr−Cretaceous, Pl−Pliocene, Q−Quaternary.

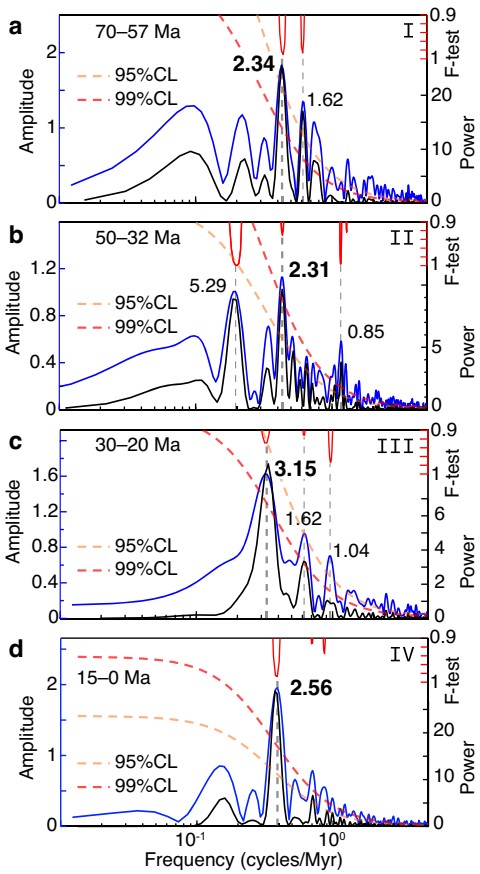

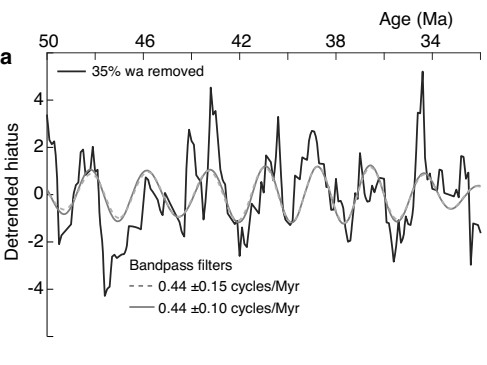

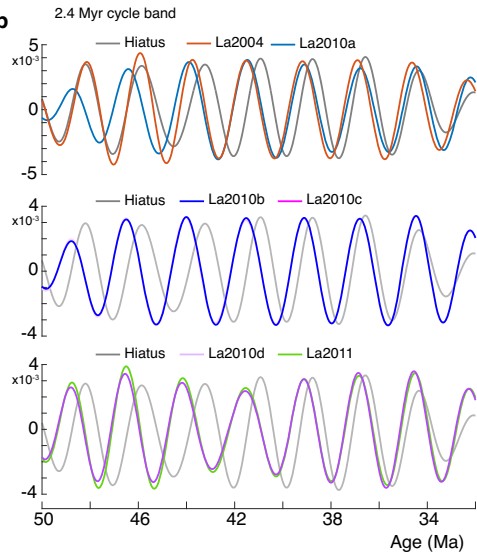

**Fig. 3 | Amplitude and power spectra of hiatus cycles. a** Amplitude (blue) and power (black) spectra for interval I (70–57 Ma) of hiatus-frequency cycle stability based respectively on the 2π-multi-taper method (MTM) and the periodogram method (see "Methods" section) of detrended data. Time windows were slightly extended to avoid edge effects. The statistical F-test is in red, and the red-noise confidence levels (CL) of 95% and 99% are in dashed, orange, and red lines, respectively (see "Methods" section). The dominant spectral peaks are highlighted in bold. **b** Amplitude and power spectra with annotations as in (**a**) for interval II (50–32 Ma). **c** Amplitude and power spectra with annotations as in (**a**) for interval III (30–20 Ma). **d** Amplitude and power spectra with annotations as in (**a**) for interval IV (15–0 Ma).

**Fig. 4 | Correlation of the hiatus-frequency and Earth's orbital eccentricity models at the 2.4 Myr cycle band, over the interval 50–32 Ma. a** Detrended hiatus-frequency data (black), and bandpass filtered data of 2.4 Myr eccentricity-related cycle (red). **b** Bandpass filters (0.44 ± 0.1 cycles/Myr) of the 2.4 Myr eccentricity cycle from astronomical solutions La2004, La2010a–d and La2011.

extremely brief and require a quantitative assessment of fragmentation of foraminifer tests[27]. These events can be linked to geologically rapid (<10 kyr) injections of $p$CO$_2$ into the atmosphere[28] resulting in a coupled decline of carbonate saturation and pH of the oceans, but they are too short to be detected as hiatuses in our dataset. On longer timescales, carbonate saturation of the ocean is regulated primarily by continental weathering, not by atmospheric $p$CO$_2$, resulting in a decoupling of carbonate saturation and pH[29]. This means that longer 'carbonate crash' events, such as the middle to late Miocene crash in the Pacific lasting ~2–3 Myr, are linked to processes unrelated to atmospheric $p$CO$_2$ and include deep-water exchange causing a

reduction in carbonate accumulation rates[30]. Therefore, our dataset, which includes only a small number of hiatuses below the CCD, reflects processes driven by deep-water modification of seafloor sediment.

We propose that maxima in the Eocene hiatus cycles correspond to orbitally-forced intensification of deep-water circulation and erosive bottom current activity, linked to eccentricity maxima which are associated with peaks in insolation and seasonality[8]. The mechanism that could link insolation cycles of shorter timescales to insolation cycles of longer timescales is amplitude modulation (AM), which is a common feature in the theoretical astronomical variations. The fundamental Milankovitch cycles, 19–23 kyr precession, 41 kyr obliquity, and 100 and 400 kyr eccentricity, are modulated by longer orbital cyclicities (e.g., 1.2 Myr, 2.4 Myr, etc.). The energy transfer from shorter to longer cycles as a result of an AM process is detected in climate and carbon-cycle sedimentary records with highly resolved proxies, where

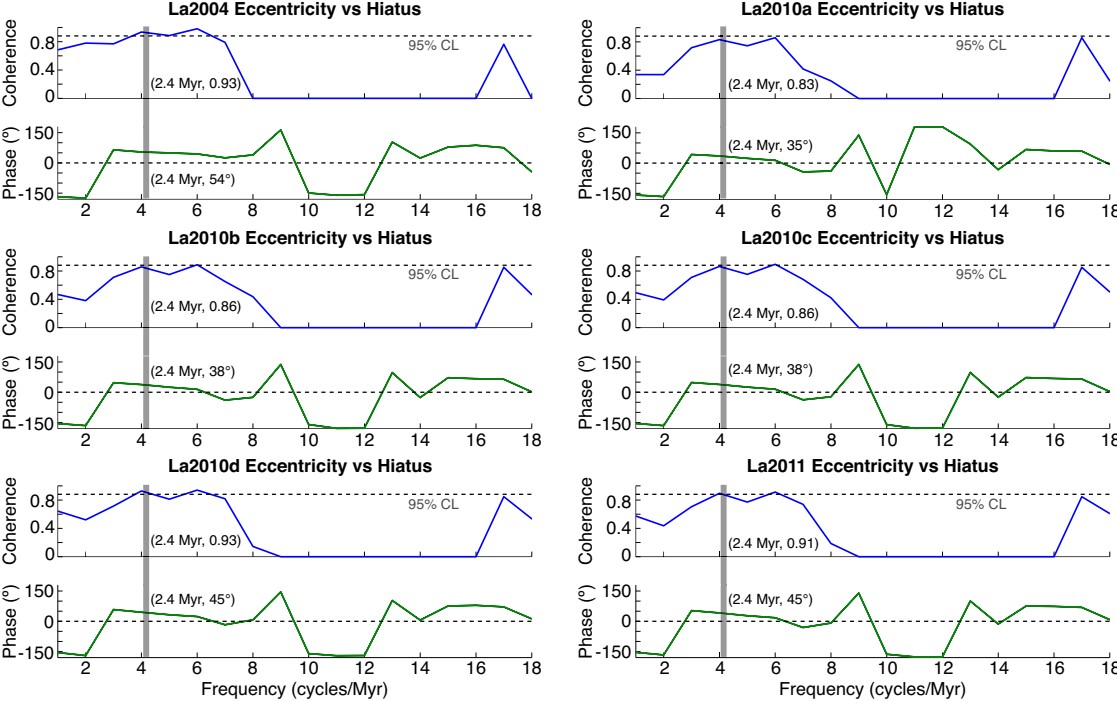

**Fig. 5 | Coherence analysis of the hiatus and eccentricity data.** 2π-MTM (multi-taper spectral method) cross coherence and phase spectra of the hiatus and La2004, La2010a–d, and La2011 eccentricity data for the interval 42–32 Ma. Note the strong coherence at the 2.4 Myr cycle band (gray bar) between the eccentricity and hiatus signals.

both high and low astronomical frequency bands were recognized (e.g., refs. 9,31). A recent general circulation model parameter sensitivity study[32] concluded that global annually-averaged surface-air temperature rises by a significant 1.75 °C when orbital eccentricity increases from 0 to Earth's maximum of 0.07[33], with more pronounced warming at the poles. Ocean modeling studies have suggested a connection between eccentricity forcing and changes in deep-ocean circulation in greenhouse climates (e.g., ref. 34). In addition, high seasonal climate variability related to eccentricity maxima has been linked to the warming of intermediate seawater[12] and increases in intensity in deep-water circulation[35]. Decadal observational data and high-resolution modeling indicate that mesoscale eddies also intensify in a warming climate system[36,37]. Eddy kinetic energy often reaches the seafloor at >5 km water depth[38], and is recognized as a driver of seafloor erosion and contourite drifts[39] where it reaches values three times higher than over the average global ocean[40]. The Drake Passage sedimentary record over the last 140 kyr suggests that strengthened westerly winds during warm climates resulted in a stronger Antarctic Circumpolar Current (ACC) and higher bottom current velocities[41]. Recent high-resolution simulations of paleo-ocean currents following the opening of circum-Antarctic gateways point to a long-term weakening of gyres associated with progressive global cooling[42], suggesting that deep-reaching eddy-induced hiatuses should also be expected to decline as a consequence. The long-term decline in deep-sea hiatuses since the middle Miocene (Fig. 2a) has been linked to late Cenozoic cooling and reduced vigour of abyssal circulation[19], lending further support to enhanced seafloor erosion during warmer climates.

We argue that the 2.34 Myr hiatus cycles over the 70–57 Ma greenhouse interval are also paced by the 2.4 Myr eccentricity forcing via the same combination of climate-dependent oceanographic mechanisms. The regularity of the cycles suggests the absence of a chaotic transition over this time interval and may provide geological constraints on orbital solutions that are highly uncertain beyond ~55 Ma[6]. The longer 3.15 Myr and 2.56 Myr hiatus cycles since ~34 Ma (Fig. 2d) have likely been augmented by non-orbital forcings

amplifying the removal of seafloor sediments. In particular, the relatively short (~10 Myr) Oligocene interval recording the 3.15 Myr cycles is bounded by two bifurcation phases (Fig. 2d) signaling instability in the climate-ocean system. Westerhold et al.[43] showed that astronomically paced climate cycles are less predictable during the Oligocene to present icehouse climate compared to warmer climates, suggesting increasingly complex, stochastic climate dynamics. Even though this interpretation is based on shorter Milankovitch cycles, their results are consistent with both the Oligocene and Miocene hiatus cycle frequency bifurcations we observe, as well as the divergence between orbital forcing and hiatus cycle frequencies.

## Hiatus-frequency bifurcations
Cycle bifurcations are transitions from a unimodal to a bimodal signal caused by a change in the dynamic state of a system[44]. Cycle bifurcations can equally arise from the modulation of close high-frequency cycles by low-frequency cycles, which is a common feature in Milankovitch astronomical cyclicities[9,31]. We suggest that bifurcations in the hiatus signal represent disturbance of the background 2.4 Myr eccentricity cycles caused by major changes in ocean circulation. These changes are chiefly driven by the opening and closing of ocean gateways modifying ocean circulation and regional patterns of seafloor erosion and hiatus formation. The bifurcation interval at ~57–52 Ma (Fig. 2d) coincides with a severe narrowing of the Norwegian-Greenland Seaway between ~56 and 53 Ma that may have altered the Atlantic deep-ocean circulation[45]. The Earth also experienced a series of short-lived but frequent extreme global warming events[46] exemplified by the Palaeocene–Eocene Thermal Maximum (PETM)[47] during which Earth's hothouse climate was more stochastic compared to a warmhouse climate due to its nonlinear response to astronomical forcing[43].

Astronomical models of the orbital motion of the inner planets predict frequent chaotic resonance transitions from libration to circulation ~65 to ~45 Ma[5,15,20,21]. These models show various possibilities in terms of their predicted resonance passages over the

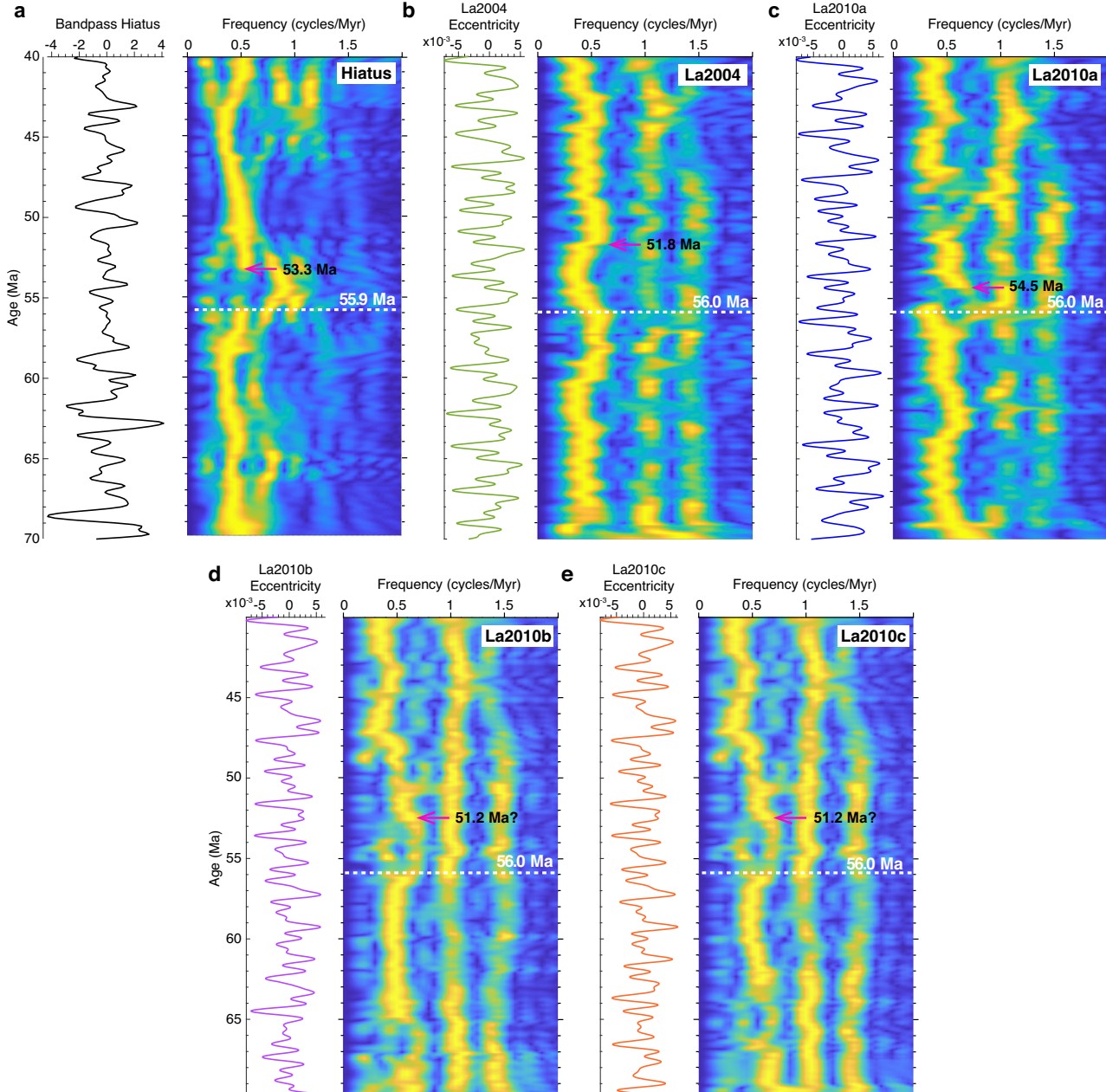

**Fig. 6 | Analysis of low-frequency variations in hiatus-frequency and Earth's orbital eccentricity data. a** Taner bandpass of hiatus-frequency data (cutoff frequencies: 0.28 and 1.8 cycles/Myr, Taner filter roll-off = 10²⁰), along with evolutive FFT spectrogram (window = 6 Myr, step = 0.1 Myr). **b–e** Taner low-pass La2004 and La2010a,b,c eccentricity data (cutoff frequencies: 0 and 1.8 cycles/Myr, Taner filter roll-off = 10²⁰), along with evolutive FFT spectrogram (window = 6 Myr, step = 0.1 Myr). **c** Taner low-pass La2004 eccentricity data (cutoff frequencies: 0 and 1.8 cycles/Myr, Taner filter roll-off = 10²⁰), along with evolutive FFT spectrogram (window = 6 Myr, step = 0.1 Myr). Horizontal, dashed-white lines indicate potential chaotic excursions, while pink arrows indicate the return to a new stable state for

the longer excursions (this does not concern the shorter excursions because their time occurrence is brief). Note the common chaotic excursion at nearly 56 Ma among the La2004 and La2010a,b,c models, matching a perturbation in the record of g4−g3 (Mars−Earth) eccentricity-related cycle in the hiatus-frequency data. In the hiatus data there is a potential shift in g4−g3 orbital term from a dominant period around 2 Myr into a dominant period around 1 Myr (see Panel **a**). For a detailed interpretation of all potential chaotic transitions within the studied 70–40 Ma time interval and using more astronomical solutions see Supplementary Figs. S1, S2 and S3.

40–70 Ma interval[6], with the nominal La2010a astronomical solution showing frequent resonances over this time interval, in contrast with other models[6]. As astronomical model predictions are uncertain, they require geological confirmation[4,14]. We test possible observable resonance passages from our hiatus-frequency data through local perturbation in the g4−g3 eccentricity term, and compare them with those theoretically predicted from six astronomical models.

Our evolutive FFT spectrogram of the hiatus-frequency data shows evidence for the record of g4−g3 eccentricity term (Fig. 6a), along with a potential frequency shift at -56 Ma. The evolutive FFT spectrogram of the nominal La2010a eccentricity data highlights several frequency shifts, in particular at the g4−g3 eccentricity term (- Fig. 6b, c and Supplementary Fig. S1). The most important shift is at nearly 56 Ma detected in the eccentricity. This concomitant shift in g4−g3 eccentricity frequency in the hiatus and astronomical data hints

at a very likely 56 Ma chaotic transition in the inner Solar System, expressed in the critical resonant argument $\theta = (s4-s3)-2(g4-g3)$[48].

We also show that the La2004 astronomical model[5] exhibits a significant deviation in the g4−g3 eccentricity term at 56 Ma (Fig. 5c), potentially reflecting a chaotic transition at the timing of the PETM. Although this possible chaotic transition at the PETM is better expressed in the La2010a than in the La2004 model[6], our geological hiatus record shows a better correlation, at the g4−g3 eccentricity cycle band, with the La2004 model. This reflects that the La2004 model is likely more reliable at longer timescales than the La2010 models[9,14,22,23]. Additionally, the La2010b and La2010c solutions, which were acquired differently (see "Methods" section), show a concomitant shift in g4−g3 eccentricity term at nearly 56 Ma.

The initiation of the Oligocene bifurcation, lasting from 34 to 30 Ma (Fig. 2d), coincides with the Eocene-Oligocene Transition, which marks a tipping-point in Earth's climate system from a greenhouse to an icehouse state[43]. Combined opening and deepening of the Drake Passage, and the deepening of the Tasman Gateway, connected all major oceans, profoundly affecting global ocean circulation and instigating the ACC[49]. Enhanced erosion of the seafloor is recorded as an increase in hiatus frequencies in the South Atlantic, Indian, and South Pacific oceans[19], and widespread contourite drifts[40]. Increased vigour of deep-sea currents and surface eddies during this time likely explains the lengthening of the hiatus cycles through excess removal of seafloor sediment in addition to orbital forcing. The following phase of relative stability lasted only 8 Myr before the onset of the following Miocene bifurcation starting at ~22 Ma (Fig. 2d), which we also attribute to major oceanographic changes superimposed on the eccentricity forcing. These include the initiation and gradual strengthening of the proto−Atlantic Meridional Overturning Circulation (AMOC) associated with the complete opening of the Drake Passage at ca. 23−17 Ma[50], the early Miocene deepening of the Fram Strait and the Greenland-Scotland Ridge[49], and the closure of the Tethys seaway at ca. 20−13.8 Ma[51]. Sensitivity experiments using a fully coupled ocean-atmosphere general circulation model have shown that progressive closure of the Tethys seaway strengthens the AMOC as well as the ACC[52]. The renewed uplift of the Greenland-Scotland Ridge between ~18 Ma and 15 Ma[49] may be at least partly responsible for extending the duration of contemporaneous bifurcation (Fig. 2d) by a temporary weakening of the AMOC.

## Chaotic orbital transition at ~56 Ma

The orbits of planets in the Solar System are not stable (chaotic) at timescales of tens to hundreds of Myr[48,53]. The chaotic diffusion in the Solar System hampers the establishment of precise astronomical solutions in deep geological time, given that uncertainties in model computation increase substantially over time[48,54]. One of the features of chaos is the emergence of frequent secular resonances in astronomical models arising especially in the inner Solar System[48,53]. The secular resonances are expressed as linear combinations of long-period orbital cycles that are detectable in the sedimentary record. Geological detection of long-period orbital cycles associated with resonant arguments allows the mapping of the timing of chaos and represents valuable constraints for astronomical modeling[15]. One of the main resonant arguments relating long orbital eccentricity period (g4−g3) and long orbital inclination period (s4−s3) is $\theta = (s4-s3)-2(g4-g3)$, which is theoretically well recognized (e.g., ref. 5), and its associated periodicities can be captured in Cenozoic sedimentary records[9,18]. The present mean periods of s4−s3 and g4−g3 are 1.2 and 2.4 Myr, respectively. However, due to chaos, these periods may deviate in the geological past, beyond 40 Ma[5,6,20]. The study of the evolution of the g4−g3 cycle in early Cenozoic strata can provide constraints on the timing of chaos and hence on astronomical modeling[5,15]. Astronomical modeling shows that the critical resonant argument $\theta$ has a present-day ratio of 2:1 between g4−g3 and s4−s3 (i.e., $s4-s3 = 2(g4-g3)$); however, when shifting from one resonance to another it has been shown that the time

span of a transition which characterizes chaos is not easily predictable. The geological record offers a good opportunity to potentially capture the chaotic transition and possibly determine its duration. Geological studies of chaos in the inner Solar System are rare, but the few existing studies have shown promising results[13–15,22]. Westerhold et al.[13] used a range of ocean drilling geological data to locate such a (transient) transition between ~55 Ma and ~52 Ma at Ocean Drilling Program (ODP) sites 1258, 1262, 1265, and 1267 from the equatorial and South Atlantic Ocean that are also included in our large global hiatus dataset (Fig. 1). Their early Eocene time interval coincides almost exactly with the bifurcation in the hiatus frequencies in our study. The hiatus data show a shift at ~56 Ma of the g4−g3 eccentricity term from a periodicity of ~2 Myr to a periodicity of ~1 Myr, followed by a return to the previous periodicity of ~2 Myr (Fig. 6). This points to a potential transient chaotic transition in the orbital motion of the inner Solar System lasting ~2.6 Myr. The timing of the transition evident in our hiatus dataset is consistent with the results of Westerhold et al.[13] who found a ~1.2 Myr periodicity between 55 Ma and 52 Ma bounded by a periodicity of ~2.4 Myr at times older than 55 Ma and younger than 52 Ma in their study. Our analysis shows evidence for a chaotic transition in the g4−g3 eccentricity term at ~56 Ma occurring in four astronomical solutions (La2004, La2010a, La2010b, and La2010c), correlating with a contemporaneous bifurcation in the hiatus data (Fig. 6).

We note that the detection of potential chaotic transitions in the geological record appear to be sensitive to the onsets of transitions, rather than their mid-ages, terminations or durations. The end of a transition is not always evident (Fig. 6 and Supplementary Figs. S1, S2, and S3), while the onset is easier to capture[22]. This observation is helpful for the comparison and correlation of different records documenting this phenomenon. It is likely that the onset of the transition found in Westerhold et al.'s study[13] corresponds to the same transition we detect in our hiatus data and in La2010b and La2010c models (Fig. 6). Although the assessment of the duration of a chaotic transition is somewhat ambiguous, our hiatus data point to a duration of ~2.6 Myr versus only ~1.5 Myr in La2010a, ~4.2 Myr in La2004, and potentially ~4.8 Myr in the La2010b and La2010c models. Despite significant advances in astronomical modeling, integrating high-precision initial conditions and possible effects from major asteroids, the time interval 50−40 Ma is still subject to uncertainties[6,20]. The validity of any astronomical model within this time interval needs to be tested by correlation with the geological record[5,6,15,21]. The chaotic transition at ~56 Ma that we infer from the hiatus-frequency data broadly matches the La2004, La2010a, La2010b, and La2010c models (Fig. 6). The detection of the chaotic transition in both our hiatus data and in Westerhold et al.'s[13] study, may provide a valuable constraint on astronomical modeling[5]. The onset of the transition coincides with the timing of PETM; however, any potential causal relationship between these phenomena requires further investigation of geological and proxy records.

Deep-sea hiatuses provide a remarkable 70 Myr-long record of ~2.4 Myr eccentricity cycles disrupted by episodes of profound tectonic, oceanographic, and climatic changes that have affected the vigor of ocean bottom currents driving deep-sea erosion. Our analysis reveals a potential chaotic transition at ~56 Ma in Earth-March orbital motions, which coincides with the timing of the PETM. The discovered link between deep-sea hiatus-frequency maxima and grand cycle eccentricity peaks, corresponding to increased insolation and seasonality, suggests that warmer climates are associated with more vigorous deep-ocean circulation. Elucidating the role of astronomical forcing in driving deep-ocean circulation is important for better understanding the future response of the ocean to global warming.

## Methods
### Hiatus dataset
Our investigation of deep-sea hiatus cyclicity through time uses the global compilation of Dutkiewicz and Müller[19], which is based on

carefully selected age-depth models for 293 deep-sea drill holes from the NSB (Neptune Sandbox Berlin) database[55]. The hiatus data used in this study are available via Zenodo at https://doi.org/10.5281/zenodo.8295655. The age-depth models are based on a combination of biostratigraphic, magnetostratigraphic, and other event data, with outlier detection[55]. Sites for which the quality of the age-depth model was ranked good to excellent by NSB with a median error ranging from about ±0.5 Myr to as low as ±0.2 Myr for orbitally-tuned sections[55] were used. The overall uncertainty of the age models based on combined Deep-Sea Drilling Project (DSDP) and Ocean Drilling Program (ODP) data in the NSB database is estimated to be ±0.37 Ma for the Cenozoic[56]. This error is sufficient for the analysis of the 2.4 Myr (g4 −g3) eccentricity cycle and the 1.2 Myr (s4−s3) obliquity cycle, and is expected to be even smaller for the more accurately dated stratigraphic sections sampled by the subsequent Integrated Ocean Drilling Program and the International Ocean Discovery Program (IODP), and further revisions of existing age-depth models[55]. Our dataset comprises 370 hiatuses defined as stratigraphic discontinuities longer than 0.18 m.y. following Spencer-Cervato[56] from all major ocean basins spanning the last 70 Myr. This conservative hiatus duration limit applies to our analysis because our dataset includes many of the holes (now revised but unchanged in terms of resolution) that Spencer-Cervato (1998) used plus additional IODP sites with higher resolution. The dataset is spread across all major ocean basins and includes sites on submerged continental crust, normal oceanic crust, LIPs, proximal and distal to ocean gateways, and at variable water depths, thus minimizing sampling bias (Fig. 1). The number of holes increases steadily from 70 Ma to the present-day. Most regions are well represented from ~60 Ma with the exception of the South Pacific, the Southern Ocean and the Mediterranean (Fig. 7a), which remain relatively unexplored compared to other regions of the global ocean. The North Pacific shows an increase in the number of holes penetrating stratigraphy younger than ~23 Ma relative to other regions, but these holes cover a disproportionately large area of the global ocean. The slight increase in the number of holes at ~25 Ma in the North Atlantic reflects seafloor erosion and contourite drift deposition during the Oligocene and Miocene[40]. The mode of hiatus duration is relatively uniform for the entire time series indicating the absence of hiatus

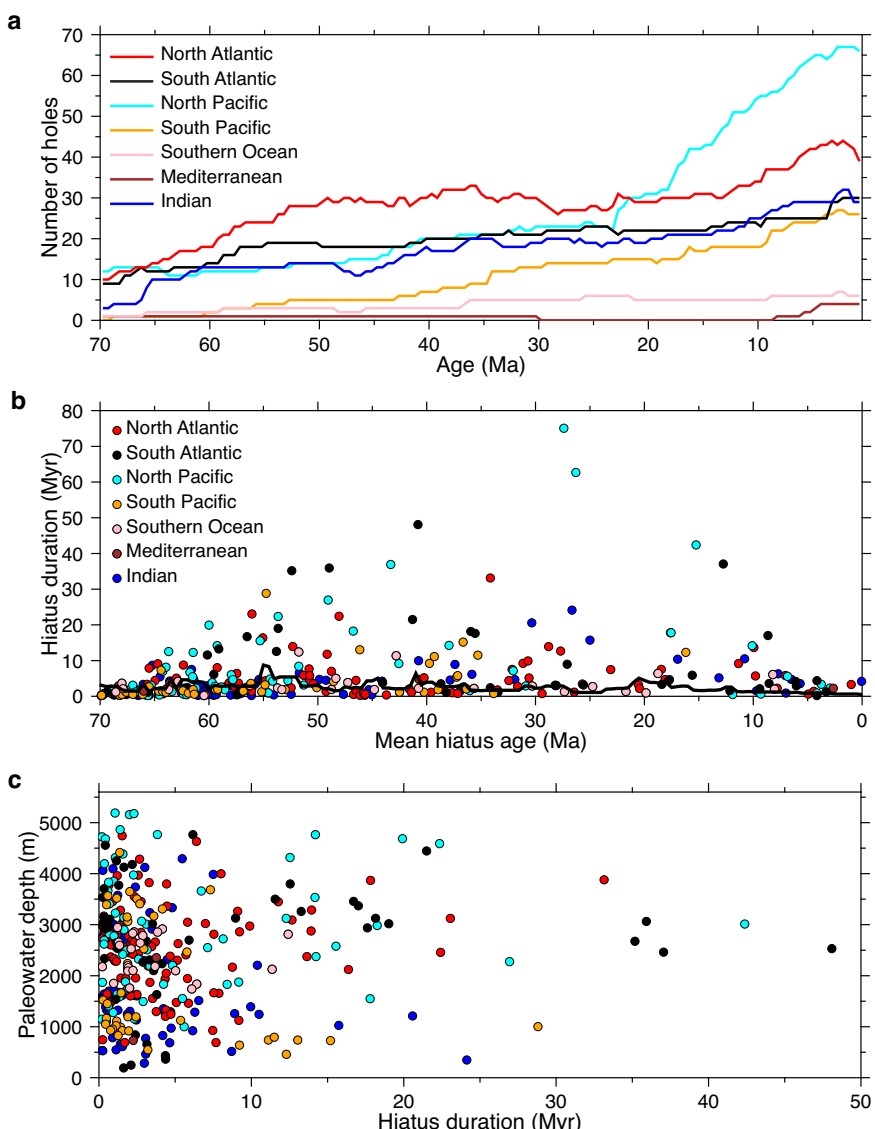

**Fig. 7 | Characteristics of all deep-sea hiatus and drill hole distributions used in the analysis. a** Number of holes drilled versus age of section penetrated. **b** Hiatus duration versus mean hiatus age. Black line represents the mode of hiatus duration based on a 5 Myr moving window and a robust maximum likelihood probability filter with outlier detection. **c** Backtracked hiatus paleo-water depth[19] versus hiatus duration showing that the majority of hiatuses are shorter than 5 Myr and occur across a wide range of paleo-water depths. Legend as in (**b**).

duration bias even for the oldest sections (Fig. 7b). This is consistent with empirical data suggesting that the effect of detecting more longer hiatuses with increasing time span is weakest in deep-water facies compared to facies on continental shelves[57]. The hiatus dataset spans a wide range of reconstructed paleo-water depths (Fig. 7c) with roughly 15% of holes located on elevated ocean floor associated with a major LIP (Fig. 1). All age-depth models were converted to the Geological Timescale 2020[58]. Hiatuses were recognized as zero-slope intervals in each age-depth model[59]. The occurrence of hiatuses or their absence (i.e., the presence of conformities) was sampled at relatively dense intervals of 100 kyr to avoid bias in the detection of the onset or cessation of hiatuses. We note that the age of hiatus onset is a maximum constrained by the chronology of the underlying sediment considering that the volume of sediment removed is unknown[60], while the age of hiatus termination is based on the chronology of overlying sediment and is more precise. A priori it is unknown what the combined effect of uncertainties in the age models and the volumes of missing sediment for individual sites is. Our approach of a big data analysis, combining a large number of sites widely distributed geographically and in terms of water depth, is designed to average out biases which may be present at individual sites.

## Spectral analysis

Long-term irregular trends in the hiatus data were measured and removed using the loess weighted average[61]. For spectral analysis, we used the multi-taper spectral method (MTM) associated with the harmonic F-test, with three $2\pi$ prolate tapers[62]. The F-test was used to seek evidence for individual lines within frequency bands with elevated amplitude that could be related to specific significant cyclicities. The MTM and F-test results were then compared to those of the basic Fourier transform squared Periodogram (unsmoothed periodogram, Matlab's periodogram.m) along with the classical autoregressive AR (1) red-noise model. We used the Gaussian band-pass filter to extract the targeted cycles[63]. Additionally, we used the Fast Fourier Transform (FFT) amplitude spectrogram to study the evolution of cyclicities through time in the hiatus-frequency and Earth's orbital eccentricity data[5,6,20]. The FFT spectrogram is based on a function computing a running periodogram of a uniformly sampled time series using FFTs of zero-padded segments, and normalized to the highest amplitudes (e.g., ref. [64]). Finally, correlation of cyclicities between the hiatus and eccentricity signals was quantified using the $2\pi$-MTM cross-spectral analysis provided in the Matlab *cmtm.m* routine[65]. We focused on the interval 42–32 Ma where the hiatus and eccentricity data share close 2.3 Myr cycles, and the reliability of the astronomical signal within the 42–32 Ma interval is guaranteed[5,6]. Six astronomical solutions, which were derived differently, have been analyzed: La2004[5], La2010a, La2010b, La2010c, La2010d[6], and La2011[20] (Table S1). The main difference between the La2004 model and the La2010a–d and La2011 models is that the La2004 initial conditions are adjusted to the JPL (Jet Propulsion Laboratory) numerical ephemeris DE406[66] over −5000 yr to +1000 yr from the present date, while La2010a–d, La2011 are fitted to the high-precision planetary ephemeris INPOP[67–69] (INPOP: Intégration Numérique Planétaire de l'Observatoire de Paris) over longer time intervals. La2010a and La2010b are fitted to INPOP08a[68] over 0.58 Myr time interval with a smaller step size of the numerical integration ($10^{-3}$) for La2010a and a larger step size of the numerical integration ($5 \times 10^{-3}$) for La2010b. The La2010a,b solutions include the effects from five major asteroids (Ceres, Vesta, Pallas, Iris, and Bamberga)[20]. La2010c is fitted to INPOP08a over 1 Myr and with a larger step size of the numerical integration ($5 \times 10^{-3}$), but does not include the five major asteroids. La2010d is fitted to INPOP06[69] over 1 Myr with a larger step size of the numerical integration ($5 \times 10^{-3}$), and including the five major asteroids. La2011 is fitted to INPOP10a (Fienga et al., 2011) over 1 Myr, which in addition includes the five major asteroids.

## Data availability

The hiatus data used in this study are available via Zenodo at https://doi.org/10.5281/zenodo.8295655.

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

## Acknowledgements

This research was supported by the Australian Research Council Future Fellowship grant FT190100829 to A.D. S.B. was supported by the French Agence Nationale de la Recherche (Grant ANR-19-CE31-0002-01) and the European Research Council under the European Union's Horizon 2020 research and innovation program (Advanced Grant AstroGeo-885250).

## Author contributions

A.D. and R.D.M. conceived the initial idea for the study and prepared the datasets. S.B. performed the spectral analysis. A.D., S.B., and R.D.M. contributed to the interpretation of the results. A.D. wrote the manuscript with input from S.B. and R.D.M.

## Competing interests

The authors declare no competing interests.
