## [Peer Review File · Nature Communications]

Deep-sea hiatus record reveals orbital pacing by 2.4 Myr
eccentricity grand cyclesREVIEWER COMMENTS

Reviewer #1 (Remarks to the Author):

Deep sea hiatus record reveals orbital pacing by 2.4 Myr eccentricity grand cycles
By Dutkiewicz et al.

- What are the noteworthy results? Yes.
- Will the work be of significance to the field and related fields? How does it compare to the established literature? If the work is not original, please provide relevant references. Yes, the work is significant to multiple fields and complements the literature.
- Does the work support the conclusions and claims, or is additional evidence needed? Some of the conclusions and claims are presented a little convoluted and additional clarifications are needed.
- Are there any flaws in the data analysis, interpretation and conclusions? - Do these prohibit publication or require revision? The publication uses a previously published database by the authors, which needs to be better introduced into the manuscript clarifying analysis strategy.
- Is the methodology sound? Does the work meet the expected standards in your field? Yes, however some clarifications are needed to increase readability for non-specialists.
- Is there enough detail provided in the methods for the work to be reproduced? Yes, the dataset as well as the code are available and can be reproduced.

General comments on the manuscript:

The manuscript presented by Dutkiewicz et al. investigates the sedimentary deep-sea record to pinpoint orbital forcing beyond the more commonly discussed short-period orbital cycles such as the Milankovitch cycles. The manuscript combines previous work by the lead author with detailed spectral analysis and comparison with published astronomical models.

The manuscript shows illustratively the potential to use disruptions in deep sea sedimentation to track astronomical cycles on a Myr scale, which is an innovative and novel approach to cyclostratigraphy. While reviewing the manuscript several points stood out, which should be addressed to facilitate the readers understanding and general applicability of the suggested concepts. Overall, the presentation of the analysis is written very technical and might therefore be difficult to follow for a non-specialist.

The stratigraphic database:

Overall, the manuscript relies heavily on previous work (Dutkiewicz & Müller, 2022), which provides a stratigraphic database of deep-sea hiatuses drilled by scientific ocean drilling. The main text of the manuscript is giving very little information on this database and the methods chapter in the appendix offers some information on the uncertainties invoked within the age constraints, but offers little information on the selection process, such as which sites were used and what criteria have been used to identify the hiatuses presented. For a full understanding of the dataset used, the reader needs to refer to the previous publication, which has a different focus, so questions raised by the cyclostratigraphic analysis might not be fully explored in the companion manuscript.

- Does the database inherit any biases (e.g. more datasets close to gateways, on elevated ocean floor (Large Igneous Provinces) and little to no data in large ocean basins or how many datasets are actually reporting on the deepest part of the record (70 Myr)).
- What is the procedure to ensure, that all hiatuses used are actually related to ocean current processes and no other erosion is sampled?
- "The age of the hiatus is constraint by the chronology of the underlying sediment considering that the volume of sediment removed is unknown, while the age of hiatus termination is based on the chronology of the overlying sediment and is more precise"(L. 234 – 237). Given the uncertainty of how much sediment has been eroded and potential times of non-deposition e.g. prior to the resumption of sedimentation, along with the uncertainties provided by the age model itself who much does these factors influence the analysis? This needs to be presented as a caveat of the analysis.
- What are the statistics of the observed stratigraphic discontinuities regarding their length and global

spread?

The original paper offers some of this information, but more information on the database itself is needed to provide a standalone manuscript and further support the suitability of the dataset for the analysis.

Use of astronomical models:

The authors investigate three iterations of the long-term astronomical computations by Laskar et al. for their correlations.

- How does the robustness of the computations align with the timeframe covered by the stratigraphic database? Laskar et al. point out an uncertainty beyond 40 -50 Myr, which covers one of the main timeframes investigated in this manuscript.
- La2004 seems to be the better fit for the dataset (as pointed out by other authors in previous studies). The authors mention that the difference between La2011 and La2004 is the integration of asteroid bodies into the computation. Are there any explanation, how this added complexity in the model is reducing the geological imprint of the cyclicity?

Connection of eccentricity forcing and deep-ocean circulation

The connection between eccentricity and ocean circulation is well pointed out using smaller timescales as examples.

- Thinking about the long-term eccentricity changes explored in this manuscript, are the authors expect similar orders of magnitude of change in the deep-ocean circulation caused by the grand cycles?

Hiatus frequency bifurcations

The authors highlight three phases of bifurcations or disturbance in the 2.4 Myr eccentricity cycles, which coincides with major changes in ocean circulation and/or potential chaotic resonance passages. To ease the reading experience, I suggest separating the discussion of the chaotic transition from the discussion of major tectonic events clearly by creating additional subchapters.

- The authors mention three major ocean circulation changes, which disturb or mask the 2.4 Myr cyclicity. An opening of a gateway in the polar regions (Tasman, Drake, Greenland-Scotland ridge) along with the closure of the Tethys Ocean increases ocean current activity (ACC & AMOC). The uplift of the Greenland-Scotland Ridge (18 – 15 Ma) is reducing AMOC activity but is suggested to extend the disturbance in the 2.4 Myr cyclicity. Are these changes in ocean current circulation of similar magnitude? Are there major tectonic events, which are not disturbing the signal? Could other mechanisms strengthening ocean current vigour (e.g. increased bottom water production) impact the signal?
- The suggestion to use the hiatus data and the proxy data collected by Westerhold et al. (2017) to reduce the ambiguity in astronomical models for the Palaeocene is intriguing, however the presentation of the spectrograms and analysis is not straight forward to follow in the manuscript. Are the sites used by Westerhold 2017 part of the hiatus database or are these completely independent observations? How can the tectonic signal of the deepening of the Norwegian-Greenland seaway and the signal of the astronomical transition be separated in the record? What is the difference between the La2011 model used in previous analysis and the La2010 model here? Given that La2004 seems to perform better at deeper timeslices, what difference might cause this? Figure 6 needs to be integrated better into the manuscript. The information in the caption on the chaotic excursion and the stable states is vital, but not fully integrated in the manuscript.

Figures:

The figures are all appropriate and display the data and interpretation well. For resubmission, I like to suggest to limit the use of pink and red in the same figure as well as green and red. Fig. 1 could include the location of the ocean gateways mentioned in the manuscript for quick reference.

Reviewer #2 (Remarks to the Author):

Review

Dear authors and editors,

I have read the manuscript "Deep-sea hiatus record reveals orbital pacing by 2.4 Myr eccentricity grand cycles" by Dutkiewicz and co-authors with great interest. The observation that deep-sea hiatuses may occur with a regularity is exciting and the suggestion that this may be linked to grand cycles of eccentricity is innovative. The methods and results are clearly reported. The manuscript merits publication after addressing some issues.

Main issues:

The choice of astronomical solution varies throughout the manuscript. The La2004 solution is repeatedly put forward as the solution best reflecting the changes observed from the geological record. This is not a generally accepted statement and should be toned down. The current references are cherry-picked from the available literature. As well as that, the La2004 solution is only considered reliable back to 40 Ma and this should be clearly stated.

Related to this, it is not entirely clear why for some analyses, the La2004 solution is preferred (Fig 4), and for others, the La2010 solution (Fig 6). It would be good to discuss the reasoning for the choice of astronomical solution more clearly. The fact that the data sometimes match one solution better and sometimes the other is interesting in itself and deserves more thorough discussion. The authors may consider presenting the comparisons a bit more clearly, perhaps by the addition of figures in the supplementary information that include all three solutions (La2004, La2010 and La2011) and perhaps, if relevant, the Zeebe solutions.

The words "amplitude bifurcations" are used a lot, and it is assumed that these are caused by extreme climate and tectonic events. Regular bifurcations can be interesting in time series analyses, for example as indicators of different periodicities interacting. The phenomenon here is not so much a bifurcation as a general weakening of the signal. I suggest making this clear throughout the text and to limit the use of the word bifurcation.

The tectonic events listed are not so well constrained in time as the manuscript suggests. The opening of gateways such as the Drake Passage is dated very differently by different authors. I recommend stating this clearly in the text and to give a range of ages. In figure 2 these events could be indicated with bars rather than arrows.

A potential chaotic transition is recorded in the hiatus-record and compared to Westerhold et al., 2017. This discussion may be expanded a bit, since the Eocene time scale has been under intense scrutiny, and other cyclostratigraphic studies have obtained different results. Consider including the recent paper by De Vleeschouwer et al, 2023, who discuss the g3 and g4 frequencies, and potential chaotic transitions, as reported by different authors (De Vleeschouwer, D., Penman, D.E., D'haenens, S., Wu, F., Westerhold, T., Vahlenkamp, M., Cappelli, C., Agnini, C., Kordesch, W.E., King, D.J. and Van Der Ploeg, R., 2023. North Atlantic Drift Sediments Constrain Eocene Tidal Dissipation and the Evolution of the Earth-Moon System. *Paleoceanography and Paleoclimatology*, 38(2), p.e2022PA004555).

The mechanisms invoked to explain deep-sea hiatuses are supported by observations from the geological record. However, it should be emphasized that these observations are done on very short time scales compared to the present study (kyrs rather than Myrs).

Minor points:

L 19: The phrase "The stable phase of hiatus-frequency cycles is bounded by amplitude bifurcations

caused by" is confusing by the choice of "phase" and "bounding", it suggests something is influencing the phase of the cycles. Consider rephrasing along the lines of "Intervals of stable hiatus-frequency cycles are interrupted by episodes of weakened cyclicality, which may be caused by ..."

L 73 and L 110: different ages are cited for the reliability of the orbital solutions. Explain which solutions you are referring to (L 74 needs references) and why these numbers are different.

L 126: consider changing circulations to circulation

Reviewer #3 (Remarks to the Author):

To the editors of Nature Communications,

Thanks for this opportunity to review an interesting manuscript from Dutkiewicz et al. The paper focuses on a record of the frequency of hiatuses in the marine stratigraphic record, derived from a multi-Myr compilation of deep-sea records. Dutkiewicz and co-authors perform cyclostratigraphic analyses on these data and convincingly demonstrate that they encode a ~2.4 Myr signal, roughly corresponding to a "grand cycle" modulating eccentricity variability.

The methods used in the paper are fairly ubiquitous in cyclostratigraphic literature, but the application to this type of data at such a long-term scale is novel. I think the results would be interesting to a wide audience, and they shed light both on large-scale changes in oceanography, as well as to the past motions of our Solar System.

My recommendation is that the paper be published with revisions. There are four overarching issues that I would like to see addressed:

- 1) The introductory text contains some spurious material and is a little overly flowery. I think some of the material could be tightened up quite a bit, or perhaps replaced with more relevant overview information about the links between astronomical forcing and climate.
- 2) It has been pretty well established that inferred sedimentation rate is positively correlated with the total duration of the studied section (i.e. the "Sadler Effect"). This is because of the discontinuous nature of sedimentation, meaning that you accumulate more "missing time" as the total duration of your study section increases. Something that's not addressed in Dutkiewicz et al. (or in their original 2022 study in Geology) is how this might possibly bias the record. If some time intervals tend to be represented by more short-duration sections, for example – would this impact the frequency of hiatuses?
- 3) Another potential source of bias in the data seems like it could come from sampling locations. If, for example, all of your data for a specific time interval came from the North Atlantic, that part of the hiatus curve wouldn't reflect a global signal, it would reflect a North Atlantic signal. Have the authors considered this?
- 4) The discussion section is interesting, but the authors seem to focus primarily on paleogeographic and tectonic factors. I'd like to see more consideration given to the potential role of atmospheric pCO₂ (which indeed, has been argued to be a more important driver for long-term paleoclimatic trends – cf. Pagani et al., 2005). It seems completely plausible to me that elevated pCO₂ could influence deposition, either by warming the atmosphere (and enhancing runoff – for example) or by acidifying the ocean, leading to more widespread dissolution. It seems absolutely worth a few lines of discussion.

I also had a few minor comments on individual lines and terminology below:

Line 1 (and throughout paper): A minor thing to consider: geologists often use "orbital" and "astronomical" interchangeably, but in my experience astronomers don't. Technically obliquity and precession are considered rotational parameters, not orbital parameters. I recognize this is some next-level pedantry but perhaps the authors should bear this in mind.

Line 15: The methods aren't novel, nor is the idea of analyzing multi-Myr composite datasets. The novelty is the type of data (number of sites with hiatuses). I think that should be clarified.

Line 27: I think "proving" is the wrong term here (more appropriate for math and philosophy). "Corroborating" or "confirming" would be better.

Line 28: The authors should cite Milankovitch's original work here (Milankovitch, 1941).

Line 31-32: The wording is a little ambiguous – "grand cycles" are also predicted by astronomical theory (Laskar et al., 2004).

Line 47: Couldn't a hiatus also be caused by erosion or dissolution?

Figure 2: It would be helpful to have a little more paleoclimatic context for the patterns. Perhaps the authors would consider adding a $\delta^{18}O$ curve (i.e. CENOGRID).

Thanks again for the opportunity to review this paper. If these concerns can be addressed I suggest the paper be published.

Anon.

8 Nov. 2023

Response to Reviewers' Comments

We thank the reviewers for their extensive and constructive comments on our manuscript “Deep-sea hiatus record reveals orbital pacing by 2.4 Myr eccentricity grand cycles” as well as their positive evaluation of our work. We have incorporated most of the recommendations and address all comments point-by-point below in blue text. The valuable feedback from all reviewers has led to a much improved manuscript.

Reviewer #1

Deep sea hiatus record reveals orbital pacing by 2.4 Myr eccentricity grand cycles
By Dutkiewicz et al.

- What are the noteworthy results? Yes.
- Will the work be of significance to the field and related fields? How does it compare to the established literature? If the work is not original, please provide relevant references. Yes, the work is significant to multiple fields and complements the literature.
- Does the work support the conclusions and claims, or is additional evidence needed? Some of the conclusions and claims are presented a little convoluted and additional clarifications are needed.
- Are there any flaws in the data analysis, interpretation and conclusions? - Do these prohibit publication or require revision? The publication uses a previously published database by the authors, which needs to be better introduced into the manuscript clarifying analysis strategy.
- Is the methodology sound? Does the work meet the expected standards in your field? Yes, however some clarifications are needed to increase readability for non-specialists.
- Is there enough detail provided in the methods for the work to be reproduced? Yes, the dataset as well as the code are available and can be reproduced.

General comments on the manuscript:

The manuscript presented by Dutkiewicz et al. investigates the sedimentary deep-sea record to pinpoint orbital forcing beyond the more commonly discussed short-period orbital cycles such as the Milankovitch cycles. The manuscript combines previous work by the lead author with detailed spectral analysis and comparison with published astronomical models.

The manuscript shows illustratively the potential to use disruptions in deep sea sedimentation to track astronomical cycles on a Myr scale, which is an innovative and novel approach to cyclostratigraphy. While reviewing the manuscript several points stood out, which should be addressed to facilitate the readers understanding and general applicability of the suggested concepts. Overall, the presentation of the analysis is written very technical and might therefore be difficult to follow for a non-specialist.

Thank you for the positive evaluation and for such a thorough review of our work. As detailed below, we provide a detailed description of our hiatus dataset and additional text explaining the analysis strategy. We have tried to explicitly re-write paragraphs using general formulation in order to make the presentation of analyses more accessible for non-specialists (e.g., lines 58–61, lines 75–95, lines 220–235).

The stratigraphic database:

Overall, the manuscript relies heavily on previous work (Dutkiewicz & Müller, 2022), which provides a stratigraphic database of deep-sea hiatuses drilled by scientific ocean drilling. The main text of the manuscript is giving very little information on this database and the methods chapter in the appendix offers some information on the uncertainties invoked within the age constraints, but offers little information on the selection process, such as which sites were used and what criteria have

been used to identify the hiatuses presented. For a full understanding of the dataset used, the reader needs to refer to the previous publication, which has a different focus, so questions raised by the cyclostratigraphic analysis might not be fully explored in the companion manuscript.

We have added an extensive discussion of the hiatus dataset, age constraints, uncertainties, and the minimisation of biases to the Methods section (lines 306–312 and 318–338). Please see our response to specific questions about the stratigraphic dataset below.

- Does the database inherit any biases (e.g. more datasets close to gateways, on elevated ocean floor (Large Igneous Provinces) and little to no data in large ocean basins or how many datasets are actually reporting on the deepest part of the record (70 Myr).

We provide a detailed discussion of the dataset and how the biases have been minimised in the Methods section of the manuscript (lines 321–336). Our discussion includes a new 3-panel figure (Fig. 7) showing the characteristics of all deep-sea hiatus and drill hole distributions used in the analysis, in particular the regional breakdown of hiatus distributions versus hiatus age, hiatus duration vs mean hiatus age, and paleo-water depth versus hiatus duration. Figure 1 has been modified to show the location of major LIPs as well as ocean gateways. We show that the hiatus dataset is spread across all major ocean basins and includes sites on submerged continental crust, normal oceanic crust, LIPs, proximal and distal to ocean gateways, and at variable water depths, thus minimising sampling bias.

- What is the procedure to ensure, that all hiatuses used are actually related to ocean current processes and no other erosion is sampled?

Apart from mechanical erosion hiatuses can also be caused by carbonate dissolution or non-deposition of carbonate. We have added detailed text (lines 100–115) to explain how this process has been extensively assessed based on reconstructions of hiatus paleowater-depths and regional carbonate compensation depths (CCD) — the depth at which the rate of supply of carbonate is balanced by its dissolution (Bramlette, 1961). Only a small number of hiatuses from a dozen holes may either be the result of carbonate dissolution or non-deposition of carbonate (Dutkiewicz and Müller, 2022). Dissolution events are difficult to detect stratigraphically because they can be extremely brief and require a quantitative assessment of fragmentation of foraminifer tests (Hancock and Dickens, 2006). These events can be linked to a geologically rapid (<10 Kyr) injections of $p\text{CO}_2$ into the atmosphere (Ridgwell and Schmidt, 2010) resulting in a coupled decline of carbonate saturation and pH of the oceans, but they are too short to be detected as hiatuses in our dataset. On longer time scales, carbonate saturation of the ocean is regulated primarily by continental weathering, not by atmospheric $p\text{CO}_2$, resulting in a decoupling of carbonate saturation and pH (Hönisch et al., 2012). This means that longer ‘carbonate crash’ events, such as the middle to late Miocene crash in the Pacific lasting ~2–3 Myr, are linked to processes unrelated to atmospheric $p\text{CO}_2$ and include deep-water exchange causing a reduction in carbonate accumulation rates (Lyle et al., 1995). Therefore, our dataset, which includes only a small number of hiatuses below the CCD, reflects processes driven by deep-water modification of seafloor sediment.

- “The age of the hiatus is constraint by the chronology of the underlying sediment considering that the volume of sediment removed is unknown, while the age of hiatus termination is based on the chronology of the overlying sediment and is more precise” (L. 234 – 237). Given the uncertainty of how much sediment has been eroded and potential times of non-deposition e.g. prior to the resumption of sedimentation, along with the uncertainties provided by the age model itself who much does these factors influence the analysis? This needs to be presented as a caveat of the analysis.

We have added the following text (lines 343–346): A priori it is unknown what the combined effect of uncertainties in the age models and the volumes of missing sediment for individual sites are. Our approach of a big data analysis, combining a large number of sites widely distributed geographically and in terms of water depth, is designed to average out biases which may be present in individual sites.

- What are the statistics of the observed stratigraphic discontinuities regarding their length and global spread?

We address this with the new text in the Methods section of the manuscript and in our new figure (Fig. 7), which shows hiatus distributions versus hiatus age, and hiatus duration vs mean hiatus age. We find that the mode of hiatus duration is relatively uniform for the entire time series indicating the absence of hiatus-duration bias even for the oldest sections. For further detail please see our response to Reviewer #1's comment above regarding the stratigraphic database.

The original paper offers some of this information, but more information on the database itself is needed to provide a standalone manuscript and further support the suitability of the dataset for the analysis.

We agree and have revised the manuscript accordingly as detailed above.

Use of astronomical models:

The authors investigate three iterations of the long-term astronomical computations by Laskar et al. for their correlations.

- How does the robustness of the computations align with the timeframe covered by the stratigraphic database? Laskar et al. point out an uncertainty beyond 40–50 Myr, which covers one of the main timeframes investigated in this manuscript.

We have extensively clarified this point by the analyses of six astronomical solutions (see updated Fig. 6 and new Supplementary Figures S1, S2 and S3).

Despite significant advances in astronomical modelling integrating high-precision initial conditions and possible effects from major asteroids, the time interval 40–50 Ma is still subject to uncertainties (Laskar et al., 2010a,b). It has been recommended by astronomers that the only way to test the validity of any astronomical model within this time interval is by correlation with the geological record (Laskar et al., 2011a; Laskar et al., 2011b; Laskar et al., 2004; Zeebe and Lourens, 2019; Zeebe and Lourens, 2022).

We have further clarified this point by focusing on the interval where the geological record aligns with the astronomical model, i.e. the interval 32–40 Ma (Lines 75–95), and added discussion on how the geological archives have the potential to discriminate among various proposed astronomical models (Lines 251–261).

- La2004 seems to be the better fit for the dataset (as pointed out by other authors in previous studies). The authors mention that the difference between La2011 and La2004 is the integration of asteroid bodies into the computation. Are there any explanation, how this added complexity in the model is reducing the geological imprint of the cyclicity?

Yes, La2004 seems to be more reliable for longer timescales than recent astronomical models. The second author (Slah Boulila) has discussed this with Jacques Laskar, who stated that it is possible that La2004 is more precise for long-term variations, but that La2010x should be more precise for short-term variations. Jacques Laskar added that only the geological record could say which astronomical solution is more reliable for the interval 40–50 Ma, and beyond.

Although it is stated that the inclusion of five major asteroids should severely impact on the precision of the computation of astronomical models, we now show in the updated Fig. 6 and in the new Supplementary Fig. S1 that the five major asteroids have minor effects on g4-g3 eccentricity term at timescales of several Myr (see also Lines 86–95 and the Methods section).

Connection of eccentricity forcing and deep-ocean circulation

The connection between eccentricity and ocean circulation is well pointed out using smaller timescales as examples.

- Thinking about the long-term eccentricity changes explored in this manuscript, are the authors expect similar orders of magnitude of change in the deep-ocean circulation caused by the grand cycles?

The mechanisms that could link insolation cycles of shorter timescales to insolation cycles of longer timescales is amplitude modulation (AM), which is a common feature in the theoretical astronomical variations. The fundamental Milankovitch cycles, 19-23 kyr precession, 41 kyr obliquity, and 100 and 400 kyr eccentricity, are modulated by longer orbital cyclicities (e.g., 1.2 Myr, 2.4 Myr, etc). The energy transfer from shorter (carrier signal) to longer (modulator signal) cycles as a result of AM process is detected in climate and carbon-cycle sedimentary records with highly resolved proxies, where both high and low astronomical frequency bands were recognized (e.g., Boulila, 2019; Boulila et al., 2012).

We included this text in the revised version (Lines 118–125).

Additionally, we are not in a position to assess the magnitude of change in deep ocean circulation. The magnitude of change in deep ocean circulation can only be quantitatively discerned from high-resolution, eddy-resolving ocean circulation models. These models are computationally very intensive, and there are hardly any published examples of deep-time paleo-deep water circulation models of this kind. This is a frontier area of oceanographic research.

Hiatus frequency bifurcations

The authors highlight three phases of bifurcations or disturbance in the 2.4 Myr eccentricity cycles, which coincides with major changes in ocean circulation and/or potential chaotic resonance passages. To ease the reading experience, I suggest separating the discussion of the chaotic transition from the discussion of major tectonic events clearly by creating additional subchapters. To ease the reading experience, I suggest separating the discussion of the chaotic transition from the discussion of major tectonic events clearly by creating additional subchapters.

We have separated the two items, and extended the discussion on the chaotic diffusion to make it more accessible for non-specialists (as suggested above).

- The authors mention three major ocean circulation changes, which disturb or mask the 2.4 Myr cyclicity. An opening of a gateway in the polar regions (Tasman, Drake, Greenland-Scotland ridge) along with the closure of the Tethys Ocean increases ocean current activity (ACC & AMOC). The uplift of the Greenland-Scotland Ridge (18 – 15 Ma) is reducing AMOC activity but is suggested to extend the disturbance in the 2.4 Myr cyclicity. Are these changes in ocean current circulation of similar magnitude? Are there major tectonic events, which are not disturbing the signal? Could other mechanisms strengthening ocean current vigour (e.g. increased bottom water production) impact the signal?

How an individual tectonic event influences deep ocean circulation is not linear and depends on many factors such as the latitude, orientation of gateway opening, and gateway closure. Clarifying

the role of bottom water production requires high-resolution, eddy-resolving ocean circulation model coupled to a tectonic plate and paleobathymetry model. We are not aware of this kind of work being done but it would clearly be extremely valuable in quantitatively assessing the potential contribution of other mechanisms contributing towards the disruption of the hiatus cyclicity.

- The suggestion to use the hiatus data and the proxy data collected by Westerhold et al. (2017) to reduce the ambiguity in astronomical models for the Palaeocene is intriguing, however the presentation of the spectrograms and analysis is not straight forward to follow in the manuscript. Are the sites used by Westerhold 2017 part of the hiatus database or are these completely independent observations? Are the sites used by Westerhold 2017 part of the hiatus database or are these completely independent observations?

The sites used in Westerhold 2017 are part of the global hiatus database. We clarify this in lines 241–244.

How can the tectonic signal of the deepening of the Norwegian-Greenland seaway and the signal of the astronomical transition be separated in the record?

This could be done through coupled eddy-resolving climate-ocean modelling where alternative models runs would need to be set up to test the effect of Norwegian-Greenland seaway deepening versus changes in insolation (global warming/cooling) driven by astronomical transitions on the strength of deep ocean circulation, especially the vigour of deep-reaching eddies.

What is the difference between the La2011 model used in previous analysis and the La2010 model here? Given that La2004 seems to perform better at deeper timeslices, what difference might cause this? Figure 6 needs to be integrated better into the manuscript. The information in the caption on the chaotic excursion and the stable states is vital, but not fully integrated in the manuscript.

We added a paragraph explaining the differences among the used astronomical solutions La2004, La2010x and La2011 to the Methods section.

From the new analyses of more astronomical solutions (six solutions versus only three in the first version), we suggest that the differences between L2004 and La2010x-La2011 solutions arise from the initial conditions (see Methods). We added this new result (Lines 108–116) to the manuscript.

We now have fully integrated Figure 6 in the discussion, and supplemented it by three other new figures S1, S2 and S3.

Figures:

The figures are all appropriate and display the data and interpretation well. For resubmission, I like to suggest to limit the use of pink and red in the same figure as well as green and red. Fig. 1 could include the location of the ocean gateways mentioned in the manuscript for quick reference.

We have changed the colours in Figs 2, 3 and 4. Figure 1 now includes the location of major gateways.

Reviewer #2:

I have read the manuscript “Deep-sea hiatus record reveals orbital pacing by 2.4 Myr eccentricity grand cycles” by Dutkiewicz and co-authors with great interest.

The observation that deep-sea hiatuses may occur with a regularity is exciting and the suggestion

that this may be linked to grand cycles of eccentricity is innovative. The methods and results are clearly reported. The manuscript merits publication after addressing some issues.

Thank you for your positive and constructive feedback. Your comments have been extremely helpful in improving the manuscript.

Main issues:

The choice of astronomical solution varies throughout the manuscript. The La2004 solution is repeatedly put forward as the solution best reflecting the changes observed from the geological record. This is not a generally accepted statement and should be toned down. The current references are cherry-picked from the available literature. As well as that, the La2004 solution is only considered reliable back to 40 Ma and this should be clearly stated.

We have clarified several aspects concerning the astronomical solutions. First, we extended the analyses to six astronomical solutions as recommended below by Reviewer #2. Accordingly, we revised figure 6 and added three new supplementary figures S1, S2 and S3. Second, we correlated the six astronomical solutions with the hiatus data at the g4-g3 eccentricity term. We substantially revised figures 4 and 5. Finally, we showed for the first time the similarities between some astronomical solutions (lines 75–98). We show that La2004 is similar to the nominal La2010a back to ~44 Ma. We also show that La2010b and La2010c are similar over the entire 40-50 Ma interval, and demonstrate from these two solutions that the five major asteroids have no effects on the model at the timescale of several Myr. Additionally, from La2010b-La2010c comparison we show that the crucial parameter for astronomical modelling are the initial conditions, followed by the step size of the numerical integration. Comparison of La2010d and La2011 further supports the above conclusion. La2010d and La2011 are close for the interval 30-50 Ma since the two solutions were adjusted to the initial conditions INPOP06 and INPOP10a.

Related to this, it is not entirely clear why for some analyses, the La2004 solution is preferred (Fig 4), and for others, the La2010 solution (Fig 6). It would be good to discuss the reasoning for the choice of astronomical solution more clearly. The fact that the data sometimes match one solution better and sometimes the other is interesting in itself and deserves more thorough discussion. The authors may consider presenting the comparisons a bit more clearly, perhaps by the addition of figures in the supplementary information that include all three solutions (La2004, La2010 and La2011) and perhaps, if relevant, the Zeebe solutions.

Please see our response above. We extended the analyses to six astronomical solutions.

The words “amplitude bifurcations” are used a lot, and it is assumed that these are caused by extreme climate and tectonic events. Regular bifurcations can be interesting in time series analyses, for example as indicators of different periodicities interacting. The phenomenon here is not so much a bifurcation as a general weakening of the signal. I suggest making this clear throughout the text and to limit the use of the word bifurcation.

We have included this point in Lines 162–164.

The tectonic events listed are not so well constrained in time as the manuscript suggests. The opening of gateways such as the Drake Passage is dated very differently by different authors. I recommend stating this clearly in the text and to give a range of ages. In figure 2 these events could be indicated with bars rather than arrows.

We have updated the figure using bars. The figure caption and main text reflect this change.

A potential chaotic transition is recorded in the hiatus-record and compared to Westerhold et al., 2017. This discussion may be expanded a bit, since the Eocene time scale has been under intense scrutiny, and other cyclostratigraphic studies have obtained different results. Consider including the recent paper by De Vleeschouwer et al, 2023, who discuss the g3 and g4 frequencies, and potential chaotic transitions, as reported by different authors (De Vleeschouwer, D., Penman, D.E., D'haenens, S., Wu, F., Westerhold, T., Vahlenkamp, M., Cappelli, C., Agnini, C., Kordesch, W.E., King, D.J. and Van Der Ploeg, R., 2023. North Atlantic Drift Sediments Constrain Eocene Tidal Dissipation and the Evolution of the Earth-Moon System. *Paleoceanography and Paleoclimatology*, 38(2), p.e2022PA004555).

We have expanded the discussion of the potential chaotic transition substantially. However, we prefer to not include this study in the discussion as the implication of a chaotic transition at ~ 41 Ma is tenuous and has not been confirmed by any astronomical solutions or independent observational data. It is a distraction for the interpretation of our results.

The mechanisms invoked to explain deep-sea hiatuses are supported by observations from the geological record. However, it should be emphasized that these observations are done on very short time scales compared to the present study (kyrs rather than Myrs).

Please see our response to Reviewer's 1 comment.

The mechanisms that could link insolation cycles of shorter timescales to insolation cycles of longer timescales is amplitude modulation (AM), which is a common feature in the theoretical astronomical variations. The fundamental Milankovitch cycles, 19-23 kyr precession, 41 kyr obliquity, and 100 and 400 kyr eccentricity, are modulated by longer orbital cyclicities (e.g., 1.2 Myr, 2.4 Myr, etc). The energy transfer from shorter to longer cycles as a result of AM process is detected in climate and carbon-cycle sedimentary records with highly resolved proxies, where both high and low astronomical frequency bands were recognized (e.g., Boulila, 2019; Boulila et al., 2012).

We included this text in the revised version (Lines 118–125).

Additionally, contourite deposits can form on long time scales, and constrain hiatus formation over multi-million year scales. For example, North Atlantic drifts consist of individual depositional events for contourites with an average duration of ~2–3 Myrs (Parnell-Turner et al., 2015), and represent depositional sites for mechanically eroded seafloor sediments, related to regional hiatuses. Successive shallowing and deepening of the GSR on similar time scales has been related to changing vigour of deep-ocean circulation (see Fig. 6 in Straume et al., 2022).

Minor points:

L 19: The phrase “The stable phase of hiatus-frequency cycles is bounded by amplitude bifurcations caused by” is confusing by the choice of “phase” and “bounding”, it suggests something is influencing the phase of the cycles. Consider rephrasing along the lines of “Intervals of stable hiatus-frequency cycles are interrupted by episodes of weakened cyclicity, which may be caused by ...”

This has been changed as recommended.

L 73 and L 110: different ages are cited for the reliability of the orbital solutions. Explain which solutions you are referring to (L 74 needs references) and why these numbers are different.

La2004 is precise back to 40 Ma (Laskar et al., 2004) (Laskar et al., 2004), while La2010a-d and La2011 models have tried to extend the accuracy of La2004 orbital eccentricity model back from 40 Ma to about 50 Ma (Laskar et al., 2011a; Laskar et al., 2011b) (Laskar et al., 2011a, 2011b).

Nevertheless, the accuracy of La2004, La2010a-d and La2011 over the interval 40-50 Ma should be explored by correlations with the geological records (Laskar, 2020).

We revised this point in Lines 75–95 and in the Methods Section (lines 364–379).

L 126: consider changing circulations to circulation

We have changed this.

Reviewer #3:

Thanks for this opportunity to review an interesting manuscript from Dutkiewicz et al. The paper focuses on a record of the frequency of hiatuses in the marine stratigraphic record, derived from a multi-Myr compilation of deep-sea records. Dutkiewicz and co-authors perform cyclostratigraphic analyses on these data and convincingly demonstrate that they encode a ~2.4 Myr signal, roughly corresponding to a “grand cycle” modulating eccentricity variability.

The methods used in the paper are fairly ubiquitous in cyclostratigraphic literature, but the application to this type of data at such a long-term scale is novel. I think the results would be interesting to a wide audience, and they shed light both on large-scale changes in oceanography, as well as to the past motions of our Solar System.

My recommendation is that the paper be published with revisions. There are four overarching issues that I would like to see addressed:

Thank you for your constructive review and insightful comments. Your suggestions and questions have helped us improve the manuscript.

1) The introductory text contains some spurious material and is a little overly flowery. I think some of the material could be tightened up quite a bit, or perhaps replaced with more relevant overview information about the links between astronomical forcing and climate.

We have simplified some of the text and have added more emphasis on climate. The sentence on the Astronomical Time Scale has been deleted.

2) It has been pretty well established that inferred sedimentation rate is positively correlated with the total duration of the studied section (i.e. the “Sadler Effect”). This is because of the discontinuous nature of sedimentation, meaning that you accumulate more “missing time” as the total duration of your study section increases. Something that’s not addressed in Dutkiewicz et al. (or in their original 2022 study in *Geology*) is how this might possibly bias the record. If some time intervals tend to be represented by more short-duration sections, for example – would this impact the frequency of hiatuses?

We address this in lines 330–334 and using a new figure (Fig. 7) summarising the age and duration characteristic of the hiatuses in our dataset. We find that the mode of hiatus duration is relatively uniform for the entire time series indicating the absence of hiatus-duration bias even for the oldest sections. This is consistent with empirical data suggesting that the effect of detecting more longer hiatuses with increasing time span is weakest in deep-water facies compared to facies on continental shelves (Sadler, 1999).

3) Another potential source of bias in the data seems like it could come from sampling locations. If, for example, all of your data for a specific time interval came from the North Atlantic, that part of

the hiatus curve wouldn't reflect a global signal, it would reflect a North Atlantic signal. Have the authors considered this?

We discuss this in the Methods section (lines 324–330) using the new figure (Fig. 7) showing the regional breakdown of hiatus distributions versus age. Most regions are well represented from ~60 Ma with the exception of the South Pacific (Fig. 7a), which remains relatively unexplored compared to other regions of the global ocean. The North Pacific shows an increase in the number of holes penetrating stratigraphy younger than ~23 Ma relative to other regions, but these holes cover a disproportionately large area of the global ocean.

4) The discussion section is interesting, but the authors seem to focus primarily on paleogeographic and tectonic factors. I'd like to see more consideration given to the potential role of atmospheric pCO₂ (which indeed, has been argued to be a more important driver for long-term paleoclimatic trends – cf. Pagani et al., 2005). It seems completely plausible to me that elevated pCO₂ could influence deposition, either by warming the atmosphere (and enhancing runoff – for example) or by acidifying the ocean, leading to more widespread dissolution. It seems absolutely worth a few lines of discussion.

We have added the following text to lines 100–115: “An assessment of the paleo-water depth at which each hiatus in our dataset formed versus regional reconstructions of the carbonate compensation depth (CCD) — the depth at which the rate of supply of carbonate is balanced by its dissolution (Bramlette, 1961), shows that only a small number of hiatuses from a dozen holes may either be the result of carbonate dissolution or non-deposition of carbonate (Dutkiewicz and Müller, 2022). Dissolution events are difficult to detect stratigraphically because they can be extremely brief and require a quantitative assessment of fragmentation of foraminifer tests (Hancock and Dickens, 2006). These events can be linked to a geologically rapid (<10 Kyr) injections of PCO₂ into the atmosphere (Ridgwell and Schmidt, 2010) resulting in a coupled decline of carbonate saturation and pH of the oceans, but they are too short to be detected as hiatuses in our dataset. On longer time scales, carbonate saturation of the ocean is regulated primarily by continental weathering, not by atmospheric PCO₂, resulting in a decoupling of carbonate saturation and pH (Hönisch et al., 2012). This means that longer ‘carbonate crash’ events, such as the middle to late Miocene crash in the Pacific lasting ~2–3 Myr, are linked to processes unrelated to atmospheric PCO₂ and include deep-water exchange causing a reduction in carbonate accumulation rates (Lyle et al., 1995). Therefore, our dataset, which includes only a small number of hiatuses below the CCD, reflects processes driven by deep-water modification of seafloor sediment.”

Line 1 (and throughout paper): A minor thing to consider: geologists often use “orbital” and “astronomical” interchangeably, but in my experience astronomers don't. Technically obliquity and precession are considered rotational parameters, not orbital parameters. I recognize this is some next-level pedantry but perhaps the authors should bear this in mind.

We revised it throughout all the paper.

Line 15: The methods aren't novel, nor is the idea of analyzing multi-Myr composite datasets. The novelty is the type of data (number of sites with hiatuses). I think that should be clarified.

We clarify this. The sentence now reads: “We apply spectral analysis to a novel dataset of Cenozoic deep-sea hiatuses to reveal a ~2.4 Myr eccentricity signal, disrupted by episodes of major tectonic forcing.”

Line 27: I think “proving” is the wrong term here (more appropriate for math and philosophy). “Corroborating” or “confirming” would be better.

We have changed “proving” to “confirming”.

Line 28: The authors should cite Milankovitch’s original work here (Milankovitch, 1941).

We have added the Milankovitch (1941) reference.

Line 31-32: The wording is a little ambiguous – “grand cycles” are also predicted by astronomical theory (Laskar et al., 2004).

We have re-phrased this section to: “the geological record also contains signals of much longer period “grand cycles” (Olsen et al., 2019), which are predicted by astronomical theory (Laskar et al., 2004)”

Line 47: Couldn’t a hiatus also be caused by erosion or dissolution?

We provide a separate discussion on the role of dissolution (lines 100–115) in hiatus formation based on the extensive assessment of hiatus plaeowater depth and regional CCD reconstructions in Dutkiewicz & Müller (2022). The vast majority of hiatuses were caused by mechanical erosion of seafloor sediment.

Figure 2: It would be helpful to have a little more paleoclimatic context for the patterns. Perhaps the authors would consider adding a d18O curve (i.e. CENOGRID).

We had considered adding the $\delta^{18}\text{O}$ curve to Fig. 2 but found it to be unhelpful because the hiatus signal responds to bottom current activity, which is ultimately driven by eddy kinetic energy at the surface, whereas the $\delta^{18}\text{O}$ curve tracks mean deep-sea temperature and ice sheet evolution as recorded in $\delta^{18}\text{O}$ of benthic forams. We have analyzed different compilations of Cenozoic deep-sea $\delta^{18}\text{O}$ data including the CENOGRID. Unfortunately, we didn’t find a preserved 2.4 Myr eccentricity signal. For example, for the interval 32–50 Ma, where the 2.4 Myr cyclicity is preserved and prominent in the hiatus data, there are instead irregular variations of several Myr in $\delta^{18}\text{O}$ data (2.6 to 4 Myr, Fig. 1A below). Additionally, results do not show correlations with the hiatus data (Fig. 1B below). That is why we prefer to not include the $\delta^{18}\text{O}$ data.

Figure 1

References

- Boulila, S., 2019, Coupling between grand cycles and events in Earth's climate during the past 115 million years: *Scientific Reports*, v. 9, p. 327, <https://doi.org/10.1038/s41598-018-36509-7>.
- Boulila, S., Galbrun, B., Laskar, J., and Pälike, H., 2012, A ~ 9 myr cycle in Cenozoic $\delta^{13}\text{C}$ record and long-term orbital eccentricity modulation: Is there a link?: *Earth and Planetary Science Letters*, v. 317, p. 273-281, <https://doi.org/10.1016/j.epsl.2011.11.017>.
- Bramlette, M. N., 1961, Pelagic sediments, *in* Sears, M., ed., *Oceanography*, Volume 67: Washington, DC, American Association for the Advancement of Science, p. 345-366, <https://www.biodiversitylibrary.org/page/24804575>.
- Dutkiewicz, A., and Müller, R. D., 2022, Deep-sea hiatuses track the vigor of Cenozoic ocean bottom currents: *Geology*, v. 50, p. 710-715, <https://doi.org/10.1130/G49810.1>.
- Hancock, H. J. L., and Dickens, G. R., 2006, Carbonate dissolution episodes in Paleocene and Eocene sediment, Shatsky Rise, west-central Pacific: *Proceedings of the Ocean Drilling Program. Scientific Results*, 198, v. 198, p. 1-24, <http://dx.doi.org/10.2973/odp.proc.sr.198.116.2005>.
- Hönisch, B., Ridgwell, A., Schmidt, D. N., Thomas, E., Gibbs, S. J., Sluijs, A., Zeebe, R., Kump, L., Martindale, R. C., Greene, S. E., Kiessling, W., Ries, J., Zachos, J. C., Royer, D. L., Barker, S., Marchitto, T. M., Moyer, R., Pelejero, C., Ziveri, P., Foster, G. L., and Williams, B., 2012, The Geological Record of Ocean Acidification: *Science*, v. 335, p. 1058-1063, <https://doi.org/10.1126/science.1208277>.
- Laskar, J., 2020, Astrochronology, *in* Gradstein, F. M., Ogg, J. G., Schmitz, M. D., and Ogg, G. M., eds., *Geologic Time Scale 2020*, Elsevier, p. 139-158, <https://doi.org/10.1016/B978-0-12-824360-2.00004-8>.
- Laskar, J., Fienga, A., Gastineau, M., and Manche, H., 2011a, La2010: a new orbital solution for the long-term motion of the Earth: *Astronomy & Astrophysics*, v. 532, p. A89, <https://doi.org/10.1051/0004-6361/201116836>
- Laskar, J., Gastineau, M., Delisle, J.-B., Farrés, A., and Fienga, A., 2011b, Strong chaos induced by close encounters with Ceres and Vesta: *Astronomy & Astrophysics*, v. 532, p. L4, <https://doi.org/10.1051/0004-6361/201117504>.
- Laskar, J., Robutel, P., Joutel, F., Gastineau, M., Correia, A. C., and Levrard, B., 2004, A long-term numerical solution for the insolation quantities of the Earth: *Astronomy & Astrophysics*, v. 428, p. 261-285, <https://doi.org/10.1051/0004-6361:20041335>.
- Olsen, P. E., Laskar, J., Kent, D. V., Kinney, S. T., Reynolds, D. J., Sha, J., and Whiteside, J. H., 2019, Mapping solar system chaos with the Geological Orrery: *Proceedings of the National Academy of Sciences*, v. 116, p. 10664-10673, <https://doi.org/10.1073/pnas.1813901116>.
- Parnell-Turner, R., White, N. J., McCave, I. N., Henstock, T. J., Murton, B., and Jones, S. M., 2015, Architecture of North Atlantic contourite drifts modified by transient circulation of the Icelandic mantle plume: *Geochemistry, Geophysics, Geosystems*, v. 16, p. 3414-3435, <https://doi.org/10.1002/2015GC005947>.
- Ridgwell, A., and Schmidt, D. N., 2010, Past constraints on the vulnerability of marine calcifiers to massive carbon dioxide release: *Nature Geoscience*, v. 3, p. 196-200, <https://doi.org/10.1038/ngeo755>.

- Sadler, P. M., 1999, The influence of hiatuses on sediment accumulation rates, *in* Bruns, P., and Hass, H., eds., *On the Determination of Sediment Accumulation Rates*, Volume GeoResearch Forum, Vol. 5: Switzerland, Trans Tech, p. 15–40.
- Straume, E. O., Nummelin, A., Gaina, C., and Nisancioglu, K. H., 2022, Climate transition at the Eocene–Oligocene influenced by bathymetric changes to the Atlantic–Arctic oceanic gateways: *Proceedings of the National Academy of Sciences*, v. 119, p. e2115346119, <https://doi.org/10.1073/pnas.2115346119>.
- Zeebe, R. E., and Lourens, L. J., 2019, Solar System chaos and the Paleocene-Eocene boundary age constrained by geology and astronomy: *Science*, v. 365, p. 926-929, <https://doi.org/10.1126/science.aax0612>.
- Zeebe, R. E., and Lourens, L. J., 2022, Geologically constrained astronomical solutions for the Cenozoic era: *Earth and Planetary Science Letters*, v. 592, p. 117595, <https://doi.org/10.1016/j.epsl.2022.117595>.

REVIEWERS' COMMENTS

Reviewer #1 (Remarks to the Author):

The authors have responded well to my raised questions and I have only minimal follow-up questions and suggestions.

My main points of inquiry were:

- The stratigraphic database

This is now a lot clearer and better constraint in the manuscript. Especially Fig. 7 and the modifications on Fig. 1 give a better overview on the available dataset. I like to suggest colouring the data shown in the lower 2 panels of Fig. 7 by the colours introduced in the uppermost panel for the different ocean basins. The authors are commenting on the different coverage of the oceanic basins and any inherited biases. A visualization based on the different ocean basins could highlight this portion and potentially reveal additional caveats to the reader. Overall, I think that the database has an inherited bias primarily given by how the sites for scientific ocean drilling from a paleoceanographic point of view are chosen, given that the most complete record is targeted rather than a hiatus-prone one. However, this is not a criticism, but rather a support, that this analysis is worthwhile and valid.

- Use of astronomical models

The questions raised are further clarified and explored. To facilitate the overview on the different models, it might be useful to include a table showing the specific strength and weaknesses of the chosen model.

- Connection of eccentricity forcing and deep-ocean circulation

No further comment

- Hiatus frequency bifurcations

The manuscript reads a lot clearer having the two items of discussion separated. The further exploration of the PETM event in correlation with the astronomical forcing is very intriguing. I have no further objections and am looking forward to see this work published.

Reviewer #2 (Remarks to the Author):

Thank you for the careful review, my concerns and suggestions have been sufficiently addressed. Congratulations on a very nice contribution

Reviewer #3 (Remarks to the Author):

To the editors of Nature Communications,

I appreciated the chance to see this revision of the paper by Adriana Dutkeiwicz and co-authors. I have read their response to my original review and have read through the revised manuscript. I am satisfied they have addressed all my major concerns. Reading through it I only noticed a few very minor issues of style and terminology (listed below).

Line 13: I believe the term "astronomical grand cycle" would be more accurate than "grand orbital cycle" (though I'm open to being corrected on this – as long as the authors are consistent).

Lines 72-73: The repeated use of the word "critical" stuck out to me. Perhaps a different word would be more elegant?

Line 106: The "k" in "kyr" should be lower-cased, identifying the prefix "kilo"

Otherwise, I recommend the article be published. I hope I have the pleasure of reading the published version of this article before long.

Reviewer 3
16 January 2024

Response to Reviewers' Comments

We thank the reviewers for assessing our revised manuscript “Deep-sea hiatus record reveals orbital pacing by 2.4 Myr eccentricity grand cycles”, and for their positive evaluation of our revisions. We have incorporated all of the additional recommendations and address all comments point-by-point below in blue text.

Reviewer #1

The authors have responded well to my raised questions and I have only minimal follow-up questions and suggestions.

- The stratigraphic database

This is now a lot clearer and better constraint in the manuscript. Especially Fig. 7 and the modifications on Fig. 1 give a better overview on the available dataset. I like to suggest colouring the data shown in the lower 2 panels of Fig. 7 by the colours introduced in the uppermost panel for the different ocean basins.

Thank you for your time and for the positive evaluation. We have modified Fig.7 accordingly.

- Use of astronomical models

The questions raised are further clarified and explored. To facilitate the overview on the different models, it might be useful to include a table showing the specific strength and weaknesses of the chosen model.

Thank you for this suggestion. We have included this table in Supplementary Information (Table S1).

Reviewer #2:

Thank you for the careful review, my concerns and suggestions have been sufficiently addressed. Congratulations on a very nice contribution.

Thank you for reviewing our revised manuscript. We are glad that the revision addressed all the concerns and suggestions raised in the original review.

Reviewer #3:

I am satisfied they have addressed all my major concerns. Reading through it I only noticed a few very minor issues of style and terminology (listed below).

Line 13: I believe the term “astronomical grand cycle” would be more accurate than “grand orbital cycle” (though I’m open to being corrected on this – as long as the authors are consistent).

Thank you for reviewing our manuscript for the second time and for the additional suggestions. We have changed “astronomical grand cycle” to “grand orbital cycle”.

Lines 72-73: The repeated use of the word “critical” stuck out to me. Perhaps a different word would be more elegant?

We have removed the word “critical” as it is not needed.

Line 106: The “k” in “kyr” should be lower-cased, identifying the prefix “kilo”.

Thank you for picking up this typo. We have fixed it.